# The domesticated transposon protein L1TD1 associates with its ancestor L1 ORF1p to promote LINE-1 retrotransposition

**Gülnihal Kavaklioglu[1], Alexandra Podhornik[1], Terezia Vcelkova[1], Jelena Marjanovic[1], Mirjam A Beck[1], Trinh Phan-Canh[2], Theresia Mair[3], Claudia Miccolo[4], Aleksej Drino[1], Mirko Doni[4], Gerda Egger[3], Susanna Chiocca[4], Miha Modic[5,6], Christian Seiser[1]***

[1]Division of Cell and Developmental Biology, Center for Anatomy and Cell Biology, Medical University of Vienna, Vienna, Austria; [2]Medical University of Vienna, Max Perutz Labs Vienna, Campus Vienna Biocenter, Vienna, Austria; [3]Department of Pathology, Medical University of Vienna, Vienna, Austria; [4]Department of Experimental Oncology, IEO, European Institute of Oncology IRCCS, Milan, Italy; [5]National Institute of Chemistry, Ljubljana, Slovenia; [6]Dementia Research Institute at King's College London and The Francis Crick institute, London, United Kingdom

**\*For correspondence:**
christian.seiser@univie.ac.at

**Competing interest:** The authors declare that no competing interests exist.

## eLife Assessment

This **important** article reports functional interactions between L1TD1, an RNA-binding protein (RBP), and its ancestral LINE-1 retrotransposon, which is not modulated at the translational level. The evidence for the association between L1TD1 and LINE-1 ORF1p is **solid**. The work implies that the transposon-derived RNA-binding protein in the human genome can interact with the ancestral transposable element from which this protein was initially derived. This work spurs interesting questions for cancer types, where LINE1 and L1TD1 are aberrantly expressed.

**Abstract** Repression of retrotransposition is crucial for the successful fitness of a mammalian organism. The domesticated transposon protein L1TD1, derived from LINE-1 (L1) ORF1p, is an RNA-binding protein that is expressed only in some cancers and early embryogenesis. In human embryonic stem cells, it is found to be essential for maintaining pluripotency. In cancer, L1TD1 expression is highly correlative with malignancy progression and as such considered a potential prognostic factor for tumors. However, its molecular role in cancer remains largely unknown. Our findings reveal that DNA hypomethylation induces the expression of L1TD1 in HAP1 human tumor cells. L1TD1 depletion significantly modulates both the proteome and transcriptome and thereby reduces cell viability. Notably, L1TD1 associates with L1 transcripts and interacts with L1 ORF1p protein, thereby facilitating L1 retrotransposition. Our data suggest that L1TD1 collaborates with its ancestral L1 ORF1p as an RNA chaperone, ensuring the efficient retrotransposition of L1 retrotransposons, rather than directly impacting the abundance of L1TD1 targets. In this way, L1TD1 might have an important role not only during early development but also in tumorigenesis.

## Introduction

Molecular domestication of transposable elements (TEs) or TE-derived sequences give rise to novel genes and regulatory sequences in the genome contributing to both genetic and epigenetic variation in an organism (*Jangam et al., 2017*; *Miller et al., 1992*; *Smit, 1999*). TEs are key drivers of genome instability, and thus are constantly exposed to silencing mechanisms by the host. Yet, co-option of TEs thorough evolution has created so-called domesticated genes beneficial to the gene network in a wide range of organisms (*Jangam et al., 2017*; *Modzelewski et al., 2022*). One example of such domesticated gene is LINE-1-type transposase containing domain 1 (*L1TD1*), which is the only domesticated protein-coding gene that originated from LINE-1 (long interspersed element 1 [L1]) (*McLaughlin et al., 2014*). Intact L1 retrotransposons are self-propagating by a 'copy-and-paste' mechanism known as retrotransposition involving the function of two L1-encoded proteins L1 open-reading frame 1 (L1 ORF1p) and L1 open-reading frame 2 (L1 ORF2p) (*Kazazian and Moran, 2017*). L1TD1 shares sequence resemblance with L1 ORF1p, a nucleic acid chaperone protein with binding capacity to both RNA and DNA (*Martin, 2006*). L1 ORF1p contributes to the formation of L1 ribonucleoproteins (L1-RNPs), key factors for retrotransposition of L1 elements, but its co-factors are yet to be elucidated (*Wallace et al., 2008*).

L1TD1, also named embryonic stem cell-associated transcript 11 (*ECAT11*), is expressed predominantly only in early embryogenesis and germ cells and gets rapidly downregulated upon the exit from pluripotency (*Iwabuchi et al., 2011*; *Mitsui et al., 2003*). Although it is dispensable for mouse development (*Iwabuchi et al., 2011*), L1TD1 is essential for the maintenance of human pluripotency (*Emani et al., 2015*; *Närvä et al., 2012*). In the context of human embryonic stem cells (hESCs), L1TD1 regulates translation of a distinct set of mRNAs (*Jin et al., 2024*), including core pluripotency factors (*Emani et al., 2015*), and was recently shown to facilitate the dissolution of stress granules (*Jin et al., 2024*). It has been proposed that during mammalian evolution the *L1TD1* gene was under positive selection due to the function in the maintenance of pluripotency in some species (*McLaughlin et al., 2014*). Apart from early development, L1TD1 is expressed in certain tumors, including germ cell tumors and colorectal cancers. In this context, L1TD1 was identified as a possible prognostic marker for medulloblastoma (*Santos et al., 2015*) and colorectal tumors (*Chakroborty et al., 2019*), where its expression is highly correlative with malignancy progression (*Urh et al., 2021*).

DNA methyltransferase (DNMT) inhibitors are successfully used as anticancer drugs, and recent reports suggest that the lethal response of human tumor cells to these compounds is based to a large extent on the activation of an antiviral immune response to endogenous retroviral transcripts (*Chiappinelli et al., 2015*; *Roulois et al., 2015*). DNMT inhibitor treatment of non-small cell lung carcinoma (NSCLC) patient-derived cells resulted in reduced cell viability and up-regulation of L1TD1, indicating an epigenetic control of the *L1TD1* gene (*Altenberger et al., 2017*). In addition, xenograft tumors with NSCLCs overexpressing L1TD1 showed decreased tumor growth, suggesting a negative impact of L1TD1 expression on tumor viability. In contrast, L1TD1 was shown to be required for cell viability in medulloblastoma (*Santos et al., 2015*). Hence, the exact mechanistic function of this domesticated transposon protein in human tumor cells is still unrevealed.

We have previously discovered that conditional deletion of the maintenance DNA methyltransferase DNMT1 in the murine epidermis results not only in the upregulation of mobile elements, such as intracisternal A-particles (IAPs) but also in the induced expression of L1TD1 (*Beck et al., 2021*, Table S1). These findings are in accordance with the observation that inhibition of DNMT activity by aza-deoxycytidine in human NSCLCs results in the upregulation of L1TD1 (*Altenberger et al., 2017*). Based on the potential role of L11TD1 as prognostic marker, we aimed at elucidating the molecular function of the domesticated transposon protein and its potential role for the control of viability in human tumor cells. To this end, we activated L1TD1 expression by inducing DNA hypomethylation via deletion of DNMT1 in the nearly haploid human cancer cell line HAP1 (*Carette et al., 2011*). First, we found that L1TD1 expression and L1 activation correlate with local DNA hypomethylation. Next, we identified L1TD1-associated RNAs by RNA immunoprecipitation sequencing (RIP-seq), which revealed transcripts and proteins with differential expression in the absence of L1TD1 by transcriptome and proteome analyses. We showed that L1TD1 protein binds to L1-RNPs. Importantly, L1TD1 facilitated L1 retrotransposition in HAP1 cells. Our results demonstrated that upon DNA hypomethylation L1TD1 affects gene expression and cell viability, and cooperates with its ancestor protein L1 ORF1p in the control of L1 retrotransposition.

## Results

### L1TD1 expression is activated through DNA hypomethylation in HAP1 cells

To address the function of L1TD1 in human tumor cells, we generated a defined human tumor cell model. To this end, we used the nearly haploid tumor cell line HAP1 cells that allows efficient gene editing by CRISPR/Cas-9 technology (*Llargués-Sistac et al., 2023*). A HAP1 DNMT1 knockout (KO) cell line has been previously validated for loss of DNMT1 protein and significant reduction in DNA methylation (*Smits et al., 2019*). Using this cell line, we examined first the effect of DNMT1 ablation on gene expression. Transcriptome analysis revealed the deregulation of 2385 genes (log2FC >2, adjusted p-value<0.05) (*Figure 1—figure supplement 1A* and *Supplementary file 1*). The majority of deregulated genes were upregulated and included, in addition to L1TD1, genes with function in transcription and cell differentiation and genes encoding melanoma antigen gene (MAGE) proteins and KRAB domain containing proteins (*Figure 1—figure supplement 1A–C*). The expression of MAGE proteins is restricted to reproductive tissues by chromatin-associated mechanisms, including DNA methylation, and was shown to be upregulated by DNMT inhibitors (*Lian et al., 2018*; *Weon and Potts, 2015*). In the context of tumor development, MAGE proteins interact with the transcriptional master regulator and E3 ubiquitin ligase KAP1 (TRIM28), thereby inducing the degradation of tumor suppressor proteins (*Lian et al., 2018*; *Weon and Potts, 2015*). On the other hand, KAP1 interacts with KRAB domain zinc finger proteins to repress the expression of transposons by chromatin-mediated mechanisms (*Ecco et al., 2017*; *Rosspopoff and Trono, 2024*). Interestingly, a recent report shows that deletion of the histone methyltransferase SETDB1 in HAP1 cells also results in DNA hypomethylation and upregulation of a similar set of zinc finger proteins (*Kang et al., 2024*). In addition to these two gene classes, expression of the de novo methyltransferases DNMT3A and DNMT3B was significantly upregulated in the absence of DNMT1, suggesting a potential compensatory mechanism (*Figure 1—figure supplement 1A*).

Next we analyzed the effect of DNMT1 ablation on DNA methylation of the *L1TD1* gene in HAP1 cells. As shown in *Figure 1A*, DNA methylation-specific PCR (MethyLight) analysis revealed significantly reduced DNA methylation at the *L1TD1* promoter in DNMT1 KO cells. DNA hypomethylation at the *L1TD1* gene correlated with a strong induction of L1TD1 mRNA and protein expression in DNMT1 KO cells (*Figure 1B–D*). Similarly, in the absence of DNMT1 the methylation of L1 transposons was reduced and expression of *L1 ORF1p*, the ancestor of L1TD1, was induced (*Figure 1—figure supplement 2A and B*).

To analyze the role of L1TD1 in a DNMT1 null background, we ablated L1TD1 in DNMT1 KO cells by gene editing resulting in HAP1 DNMT1/L1TD1 double knockout (DKO) cells (*Figure 1C*). Based on the observations that L1TD1 is highly expressed in human germ cell tumors (*Närvä et al., 2012*), we used the ovarian cancer cell line OV-90 as a positive control (*Figure 1D and E*). In accordance with previously published data with hESCs (*Närvä et al., 2012*), indirect immunofluorescence analysis revealed that L1TD1 was preferentially localized in the cytosol of HAP1 DNMT1 KO cells and OV-90 cells (*Figure 1E*). L1TD1 was reported to localize in P-bodies in hESCs (*Närvä et al., 2012*). We detected a similar granular localization of L1TD1 in HAP1 DNMT1 KO cells and OV-90 cells (*Figure 1E*).

Loss of DNMT1 protein in the human colorectal carcinoma cells results in chromosomal defects and apoptosis (*Brown and Robertson, 2007*). HAP1 cells with DNMT1 depletion are viable but showed significantly reduced cell proliferation and increased apoptosis (cleaved caspase 3) and DNA damage (γ-H2AX) (*Figure 1F–H*). Additional deletion of L1TD1 enhanced the antiproliferative effects of DNMT1 ablation. These results suggest that L1TD1 expression is regulated by DNA methylation and its depletion affects cell viability in DNMT1-deficient HAP1 cells.

### L1TD1 binds to L1 transcripts and a subset of mRNAs

L1TD1 was previously shown to act as RNA-binding protein by binding to its own transcript (*Närvä et al., 2012*) and selected group of mRNAs under L1TD1 translational control (*Jin et al., 2024*). To determine the L1TD1 binding repertoire in the context of L1TD1- and DNMT-dependent cell viability and proliferation control, we performed RNA immunoprecipitation (RIP) followed by sequencing (RIP-seq) (*Figure 2A*). L1TD1-containing RNP complexes (L1TD1-RNPs) were immunoprecipitated in DNMT1 KO cells by an L1TD1-specific antibody. We identified 597 transcripts enriched (log2FC >2, adjusted p-value <0.05) in L1TD1-RNPs compared to the input (*Figure 2B* and *Supplementary file*

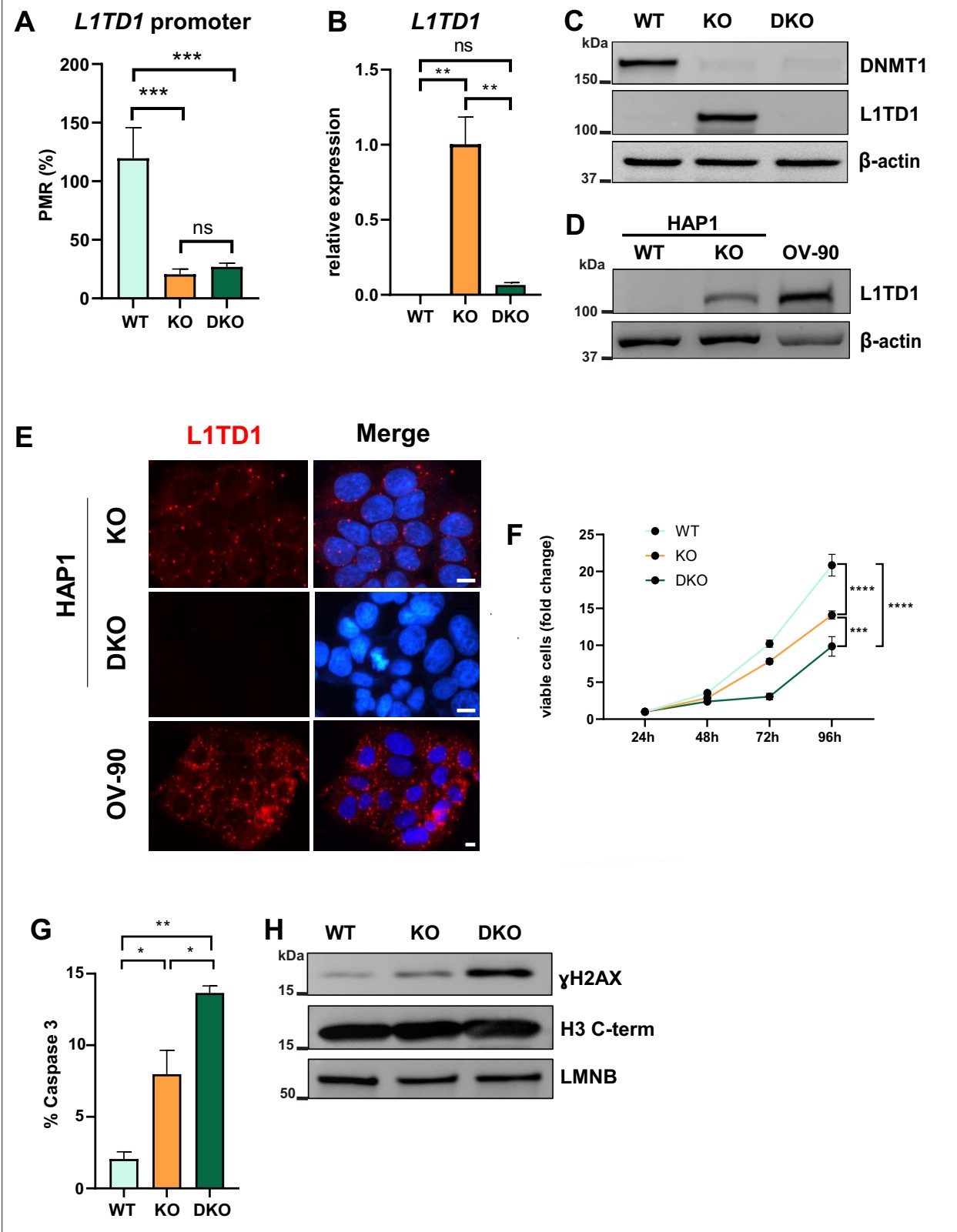

**Figure 1.** DNA hypomethylation results in the activation of L1TD1 expression and loss of L1TD1 affects cell viability in HAP1 cells. (**A**) Quantification of DNA methylation levels at the *L1TD1* promoter in HAP1 wildtype (WT), DNMT1 KO, and DNMT1/L1TD1 DKO cells using the MethyLight assay. DNA methylation is shown as percentage of methylation ratio (PMR). (**B**) qRT-PCR analysis of *L1TD1* mRNA expression in HAP1 WT, DNMT1 KO, and DNMT1/L1TD1 DKO cells. *GAPDH* was used as a normalization control and relative *L1TD1* mRNA levels in DNMT1 KO cells were set to 1. Data are shown as a

*Figure 1 continued on next page*

*Figure 1 continued*

mean of ± SD of three biological replicates. (**C**) Western blot analysis of L1TD1 levels in HAP1 WT, DNMT1 KO, and DNMT1/L1TD1 DKO cells. β-Actin was used as loading control. (**D**) Western blot analysis of L1TD1 protein expression in HAP1 WT and DNMT1 KO cells and OV-90 cells; β-actin was used as loading control. (**E**) Indirect immunofluorescence staining of L1TD1 (red) in HAP1 DNMT1 KO and DNMT1/L1TD1 DKO cells and OV-90 cells. In merged images, nuclear DNA was stained with DAPI (blue). Scale bars, 5 μm. (**F**) Cell viability analysis of HAP1 WT, DNMT1 KO, and DNMT1/L1TD1 DKO cells using the CellTiter-Glo assay measured over 96 hours (n = 6). (**G**) Bar graph representing the percentage of apoptotic cells of cultured cell lines quantified by flow cytometry analysis of cleaved caspase 3. (**H**) Western blot analysis of γH2AX levels in HAP1 WT, DNMT1 KO, and DNMT1/L1TD1 DKO cells in nuclear extracts. Antibodies specific for histone H3 C-terminus and LAMIN B (LMNB) were used as loading controls. (**A–B, F–G**) Statistical significance was determined using one-way ANOVA with Tukey's multiple comparison correction. *p≤0.05, **p≤0.01, ***p≤0.001, ****p≤0.0001, ns = not significant.

The online version of this article includes the following source data and figure supplement(s) for figure 1:

**Source data 1.** Source data for all plots in *Figure 1* and corresponding details of statistical tests and p-values.

**Source data 2.** Original western blots of *Figure 1D and H* indicating the relevant bands.

**Source data 3.** Original image files for western blots of *Figure 1D and H*.

**Figure supplement 1.** Deregulation of the HAP1 transcriptome upon loss of DNMT1.

**Figure supplement 1—source data 1.** Plots and statistics source data for all plots in *Figure 1—figure supplement 1* and corresponding details of statistical tests and p-values.

**Figure supplement 2.** DNMT1 ablation results both in DNA hypomethylation at L1 elements and expression of L1 ORF1p transcripts.

**Figure supplement 2—source data 1.** Plots and statistics source data for all plots in *Figure 1—figure supplement 2* and corresponding details of statistical tests and p-values.

*2*). In addition, we used DNMT1/L1TD1 DKO cells as a negative control. Only transcripts enriched (log2FC >2, adjusted p-value <0.05) in both assays (DNMT1 KO versus input and DNMT1 KO versus DNMT1/L1TD1 DKO) were considered as L1TD1-associated RNAs (*Figure 2—figure supplement 1A* and *Supplementary file 2*), resulting in the identification of 228 transcripts. Importantly, *L1TD1* mRNA was detected as one of the top transcripts in L1TD1-RNPs (*Figure 2B*). Of note, the transcript of *YY2*, a retrotransposon-derived paralogue of yin yang 1 (YY1) (*Tahmasebi et al., 2016*), was also enriched in L1TD1-RNPs.

To validate the RIP-seq results, selected individual transcripts were amplified by gene-specific qRT-PCR analysis after repeating the RIP. Transcripts of *L1TD1, YY2,* and *ARMC1* (as one of the RIP top hits) were specifically enriched in L1TD1-RNPs obtained from DNMT1 KO cells compared to DNMT1/L1TD1 DKO cells, whereas the negative control *GAPDH* did not show such enrichment (*Figure 2C*). Gene Ontology analyses of the 228 common hits indicated enriched groups in cell division and cell cycle regulation and, in accordance with a recent study (*Jin et al., 2024*), enriched transcripts encoding zinc finger and in particular KRAB domain proteins that have been implicated in the restriction of endogenous retroviruses (*Rosspopoff and Trono, 2023*; *Figure 2—figure supplement 1B and C*).

Analysis of RIP-Seq using TEtranscript designed for the analysis of transposon-derived sequences (*Jin et al., 2015*), in the differentially expressed sets indicated an association of L1TD1 with *L1* transcripts (*Figure 2D* and *Supplementary file 3*), which was also validated by qRT-PCR analysis of L1TD1 RIP samples (*Figure 2E*). In addition, we also identified *ERV* and *AluY* elements as L1TD1-RNP-associated transcripts (*Figure 2—figure supplement 1D* and *Supplementary file 3*). These combined results suggest that L1TD1 interacts with a set of transcripts including RNAs with functions in cell division and cell cycle control. Furthermore, L1TD1 has kept the ability to associate with *L1* transcripts potentially regulating them similar to its ancestor protein L1 ORF1p (*Wei et al., 2001*).

## Loss of L1TD1 modulates the proteome and transcriptome of HAP1 cells

To explore the cellular function of L1TD1, we analyzed the proteomes and transcriptomes of DNMT1 KO and DNMT1/L1TD1 DKO cells. A comparison of the proteomes revealed 98 upregulated and 131 downregulated proteins in DNMT1/L1TD1 DKO cells versus DNMT1 KO cells (log2FC ≥1, adj. p-value≤0.05) (*Figure 3A* and *Supplementary file 4*). Gene Ontology Enrichment Analysis revealed that differentially expressed proteins were mainly associated with regulation of cell junction and cysteine-type endopeptidase activity involved in apoptotic process (*Figure 3—figure supplement 1A*). To investigate whether the corresponding protein abundance of mRNAs contained in L1TD1-RNPs

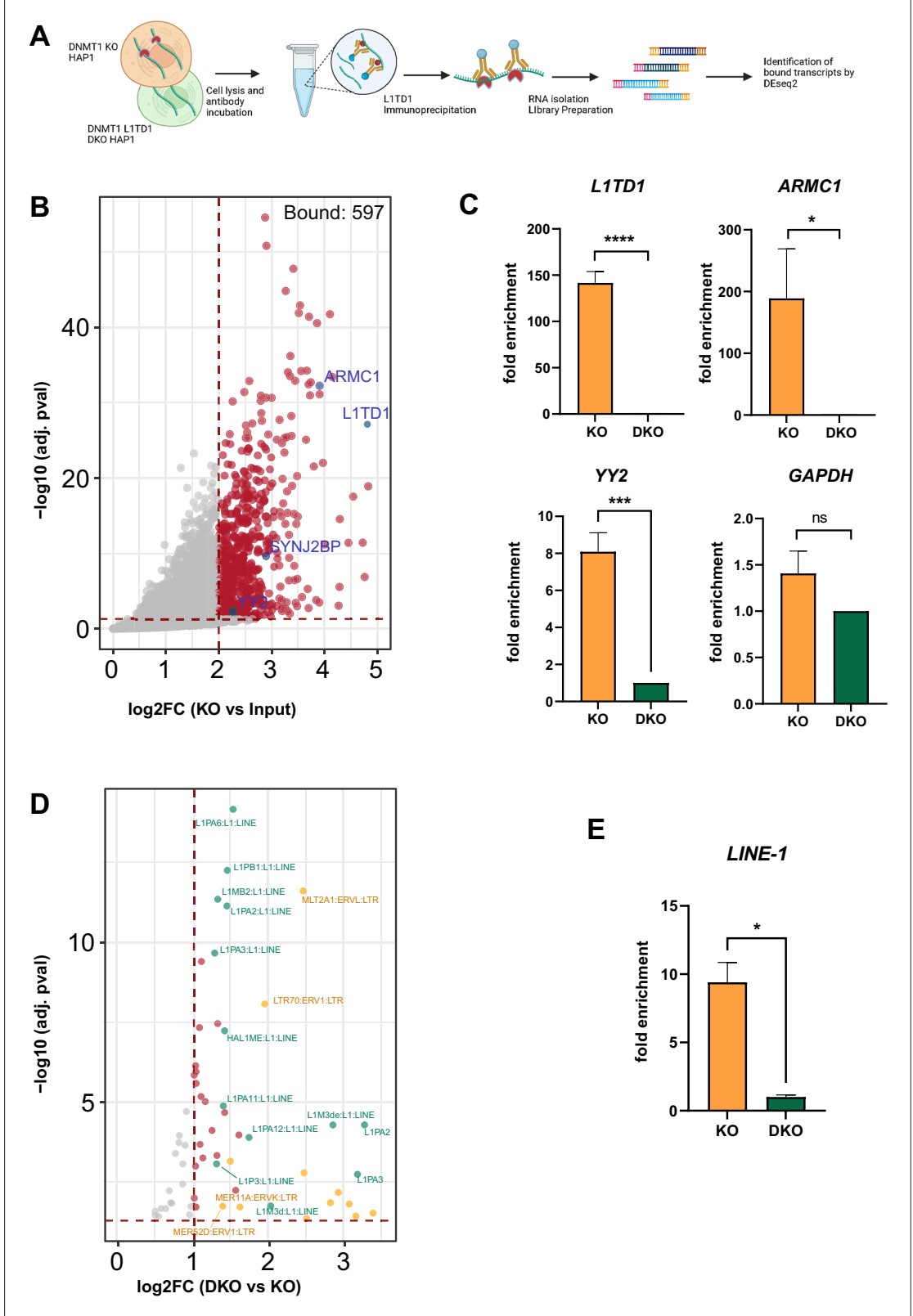

**Figure 2.** RIP-seq identifies a set of RNAs and transposon transcripts associated with L1TD1. (**A**) Schematic representation of the RNA immunoprecipitation sequencing (RIP-seq) method (figure created with BioRender.com). L1TD1-RNA complexes were isolated from HAP1 KO cell extracts with an L1TD1-specific antibody, RNA was isolated from complexes and input and cDNA libraries were prepared using the Smart-seq3 protocol. Sequencing data was analyzed by DEseq2 and TEtranscript software, separately. (**B**) Volcano plot showing L1TD1-associated transcripts as a result

*Figure 2 continued on next page*

*Figure 2 continued*

of DESeq2 analysis (cut-off log2FC >2 and adj. p-value<0.05). Selected hits are highlighted in blue. (**C**) RIP-qPCR analysis confirms L1TD1 interaction with the transcripts *L1TD1, ARMC1, YY2*. The bar graphs represent the fold enrichments of the transcripts in the IP samples of DNMT1 KO relative to DNMT1/L1TD1 DKO cells (set to 1) and normalized to input samples in the indicated cells. *GAPDH* was used as a negative control for the RIP-qPCR analysis. (**D**) Volcano plot showing L1TD1-associated transposon transcripts as result of the TEtranscript analysis with a log2FC >1 and adj. p-value<0.05. LINE1 elements are highlighted in green, ERV elements in yellow and other associated transposon transcripts in red. (**E**) RIP-qPCR analysis confirms the association of L1TD1 with *L1* transcripts. Statistical significance was determined using paired two-tailed t-test. All data in the figure are shown as a mean of ± SD of three biological replicates. *p≤0.05, **p≤0.01, ***p≤0.001, ****p≤0.0001, ns = not significant.

The online version of this article includes the following source data and figure supplement(s) for figure 2:

**Source data 1.** Source data for all plots in *Figure 2* and corresponding details of statistical tests and p-values.

**Figure supplement 1.** L1TD1 interacts with a specific set of mRNAs and transposon transcripts.

**Figure supplement 1—source data 1.** Plots and statistics source data for all plots in *Figure 2—figure supplement 1* and corresponding details of statistical tests and p-values.

is affected by L1TD1 expression, we compared proteomic data with the RIP-seq data. A comparison of differentially expressed proteins with transcripts associated with L1TD1 identified only L1TD1 (*Figure 3—figure supplement 1B*). The lack of overlap between RIP-seq and proteomics data (except L1TD1) suggested that modulation of the abundance of proteins encoded by L1TD1-associated transcripts is not the main function of L1TD1.

Interestingly, we discovered that L1 ORF1p was upregulated in the absence of L1TD1 (*Figure 3A* and *Supplementary file 4*). Western blot analysis confirmed the increased expression of L1 ORF1p in HAP1 DNMT1/L1TD1 DKO cells compared to DNMT1 KO cells (*Figure 3C*), while L1 transcript levels showed no significant change (*Figure 3—figure supplement 2A*). This indicates that L1TD1 attenuates the expression of L1 ORF1p, suggesting a potential regulatory crosstalk between L1TD1 with its ancestor protein.

To investigate a potential effect of L1TD1 on the expression levels of associated RNAs, comparative RNA-seq analysis was performed with DNMT1 KO and DNMT1/L1TD1 DKO cells. Upon loss of L1TD1, we identified 323 upregulated and 321 downregulated transcripts, respectively (log2-fold change ≥1; adjusted p-value<0.05) (*Figure 3B* and *Supplementary file 5*). About 25% of the differentially expressed proteins (34% of upregulated and 22% of downregulated proteins) identified in the proteome analysis were also significantly deregulated at the RNA level (*Supplementary file 4*). However, none of the deregulated transcripts, except *L1TD1*, was detected in L1TD1-RNPs (*Figure 3B* and *Supplementary file 2* and *Supplementary file 5*). This finding was corroborated by qRT-PCR analysis of L1TD1-RNP-associated transcripts (*Figure 3—figure supplement 2B*), suggesting that the cellular function of L1TD1 might be independent of a deregulation of its associated mRNAs.

Based on our observations implying that L1TD1 does not exert its cellular function as regulator of RNA abundance, while being associated with transposon RNAs (*Figure 2D*), we used TEtranscript (*Jin et al., 2015*) to identify differentially expressed TE-derived sequences transcriptome-wide. Loss of L1TD1 resulted in upregulation of specific SINEs such as AluY elements (*Figure 3—figure supplement 2C* and *Supplementary file 6*). Taken together, these data suggest that L1TD1 affects the abundance of transposon transcripts in DNMT1-deficient HAP1 cells.

## L1TD1 interacts with the L1 ORF1p protein

An interaction of L1TD1 with *L1* RNA (*Figure 2D*) and increased L1 ORF1p levels in L1TD1-depleted cells (*Figure 3B*) imply a direct association of L1TD1 with L1 ORF1p. We thus hypothesized that L1TD1 associates with L1-RNPs via L1 ORF1p. Therefore, we immunoprecipitated L1 ORF1p from HAP1 DNMT1 KO cells and OV-90 cells, and tested the IPs for the presence of endogenous L1TD1. Indeed, L1TD1 was co-precipitated with L1 ORF1p from DNMT1 KO cells and OV-90 cells (*Figure 3D*, *Figure 3—figure supplement 3B*). Next, we asked whether the association of L1TD1 with L1 ORF1p is mediated by RNA. As shown in *Figure 3—figure supplement 3A and B*, L1TD1 associates with L1 ORF1p in an RNA-independent manner. In agreement with their association in protein complexes, indirect immunofluorescence experiments revealed a partial co-localization of L1TD1 and L1 ORF1p in both cell lines (*Figure 3E*, *Figure 3—figure supplement 3C*). In summary, our results suggest that L1TD1 associates with its ancestor ORF1p and this association does not depend on RNA intermediates.

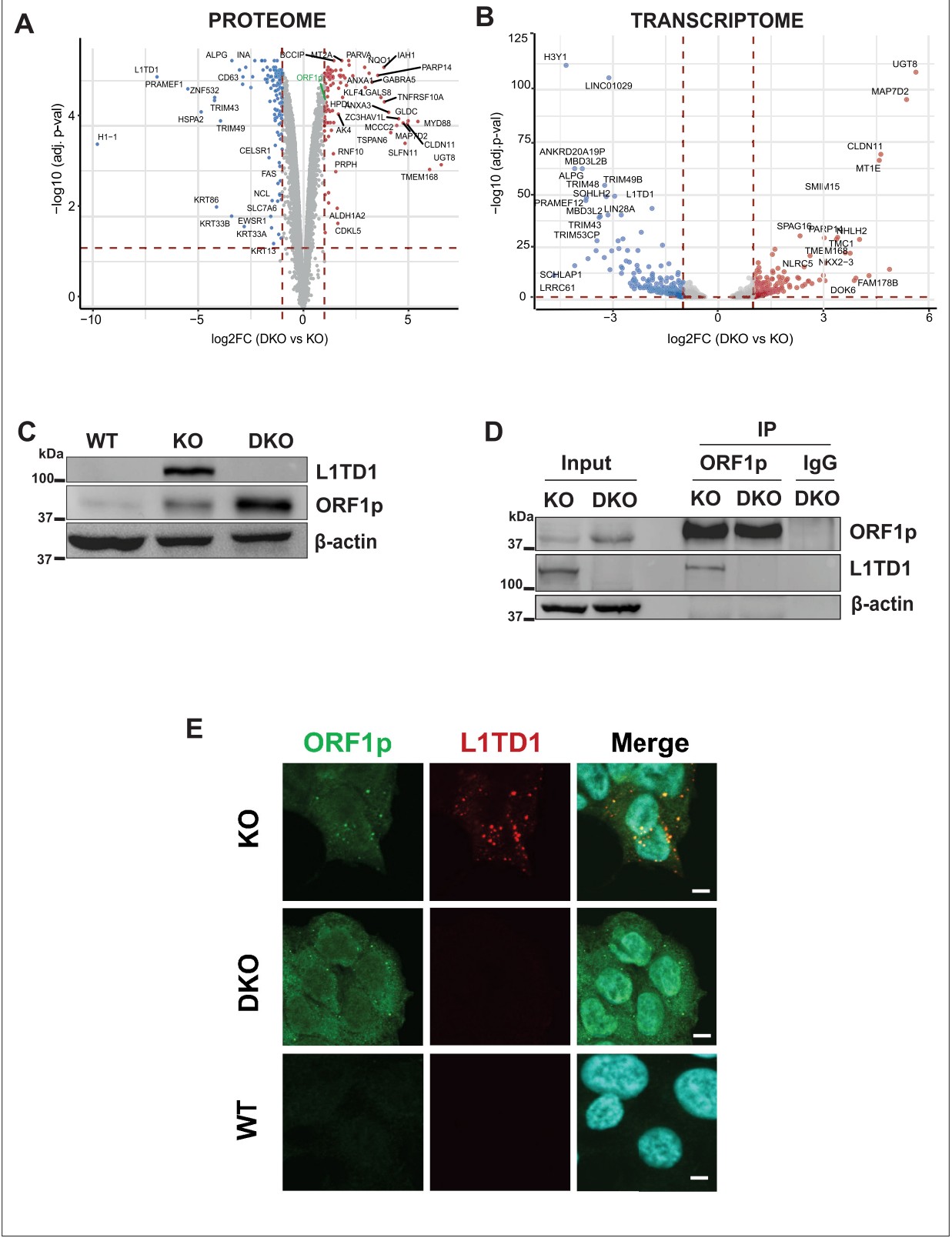

**Figure 3.** L1TD1 cross-talk with its ancestor L1 ORF1p. (**A**) Volcano plot displaying the comparison of the proteomes of HAP1 DNMT1 KO and DNMT1/L1TD1 DKO cells determined by mass spectrometry. Differentially abundant proteins were plotted as DNMT1/L1TD1 DKO over DNMT1 KO (log2FC ≥1, adj. p-value<0.05 [red] and log2FC ≤ −1, adj. p-value<0.05 [blue]). (**B**) Volcano plot illustrating the DESeq2 analysis of RNA-seq performed with HAP1 DNMT1 KO and DNMT1/L1TD1 DKO cells. Differentially expressed genes are plotted as DNMT1/L1TD1 DKO over DNMT1 KO (log2FC ≥1,

*Figure 3 continued on next page*

*Figure 3 continued*

adj. p-value<0.05 [red] and log2FC ≤ –1, adj. p-value<0.05 [blue]). (**C**) Western blot analysis illustrating protein levels of L1TD1 and L1 ORF1p in HAP1 WT, DNMT1 KO and DNMT1/L1TD1 DKO cells. β-actin was used as loading control. (**D**) Physical interaction of L1 ORF1p and L1TD1. L1 ORF1p was immunoprecipitated with an L1 ORF1p-specific antibody from whole cell extracts prepared from DNMT1 KO and DNMT1/L1TD1 DKO cells. Precipitated L1 RNP complexes and inputs were analyzed on a Western blot. IgG was used as a negative IP control and β-actin was used as loading control. (**E**) Confocal microscopy images of indirect immunofluorescence co-stainings using mouse L1 ORF1p (green) and rabbit L1TD1 (red) antibodies in DNMT1 KO, DNMT1/L1TD1 DKO and HAP1 WT cells. In merged images nuclear DNA was stained with DAPI. Scale bars, 5 µm.

The online version of this article includes the following source data and figure supplement(s) for figure 3:

**Source data 1.** Source data for all plots in *Figure 3* and corresponding details of statistical tests and p-values.

**Source data 2.** Original western blots of *Figure 3C and D* indicating the relevant bands.

**Source data 3.** Original image files for western blots of *Figure 3C and D*.

**Figure supplement 1.** GSEA analysis of proteins upon loss of L1TD1.

**Figure supplement 1—source data 1.** Source data for all plots in *Figure 3—figure supplement 1* and corresponding details of statistical tests and p-values.

**Figure supplement 2.** Ablation of L1TD1 leads to changes in the transcriptome of HAP1 cells.

**Figure supplement 2—source data 1.** Source data for all plots in *Figure 3—figure supplement 2* and corresponding details of statistical tests and p-values.

**Figure supplement 3.** RNA-independent interaction of L1TD1 and L1 ORF1p in HAP1 and OV-90 cells.

**Figure supplement 3—source data 1.** Original western blots of *Figure 3—figure supplement 3B* indicating the relevant bands.

**Figure supplement 3—source data 2.** Original image files for western blots of *Figure 3—figure supplement 3B*.

## L1TD1 has a positive impact on L1 retrotransposition

Since L1TD1 associates with *L1* RNA and modulates L1 ORF1p expression, we next asked whether the domesticated transposon protein can impact L1 retrotransposition in HAP1 cells. To this end, we carried out plasmid-based retrotransposition assays (*Kopera et al., 2016*) in DNMT1 KO and DNMT1/L1TD1 DKO cells (*Figure 4A and B*). Following transfection and blasticidin selection, the rate of retrotransposition of the reporter construct was assessed through counting blasticidin-resistant colonies, transfected either with a retrotransposition-competent reporter construct or a retrotransposition-deficient control. To take into account differences in proliferation and blasticidin sensitivity, DNMT1 KO and DNMT1/L1TD1 DKO cells were transfected in parallel with the blasticidin resistance gene vector pLenti6.2 and the numbers of clones from the retrotransposition assay were corrected for differences in transfection efficiency (*Figure 4—figure supplement 1B*). An average of 506 retrotransposition events were observed in DNMT1 KO cells in three independent experiments whereas DNMT1/L1TD1 DKO cells yielded over fivefold fewer colonies when transfected with the retrotransposition-competent reporter construct (86 retrotransposition events on average) (*Figure 4C and D*, *Figure 4—figure supplement 1A*). No blasticidin-resistant clones were obtained for the backbone vector pCEP4, but comparable effects of L1TD1 on retrotransposition rates were observed for retrotransposition-competent reporter constructs in three independent retrotransposition assays (*Figure 4—figure supplement 1A and B*). Combined, these data suggest that L1TD1 enhances L1 retrotransposition in DNMT1-deficient HAP1 cells and this effect is mediated through the association with L1 ORF1p and *L1* RNA.

## Discussion

In this study, we show that the domesticated transposon protein L1TD1 promotes L1 retrotransposition. Numerous cellular factors have been shown in the past to affect retrotransposition (*Ariumi, 2016*; *Goodier, 2016*; *Pizarro and Cristofari, 2016*; *Protasova et al., 2021*). However, many of these proteins are part of host defense mechanisms and restrict retrotransposition, and it was speculated that L1TD1 also limits L1 retrotransposition (*McLaughlin et al., 2014*). Unexpectedly, we identified L1TD1 as one of the cellular factors that has a positive impact on L1 retrotransposition, most likely by interacting with L1 RNPs and acting as RNA chaperone. L1TD1 deletion resulted in reduced L1 retrotransposition despite upregulation of L1 ORF1p, indicating that the higher levels of the transposon protein cannot fully compensate the loss of L1TD1. Interestingly, a similar effect has been previously

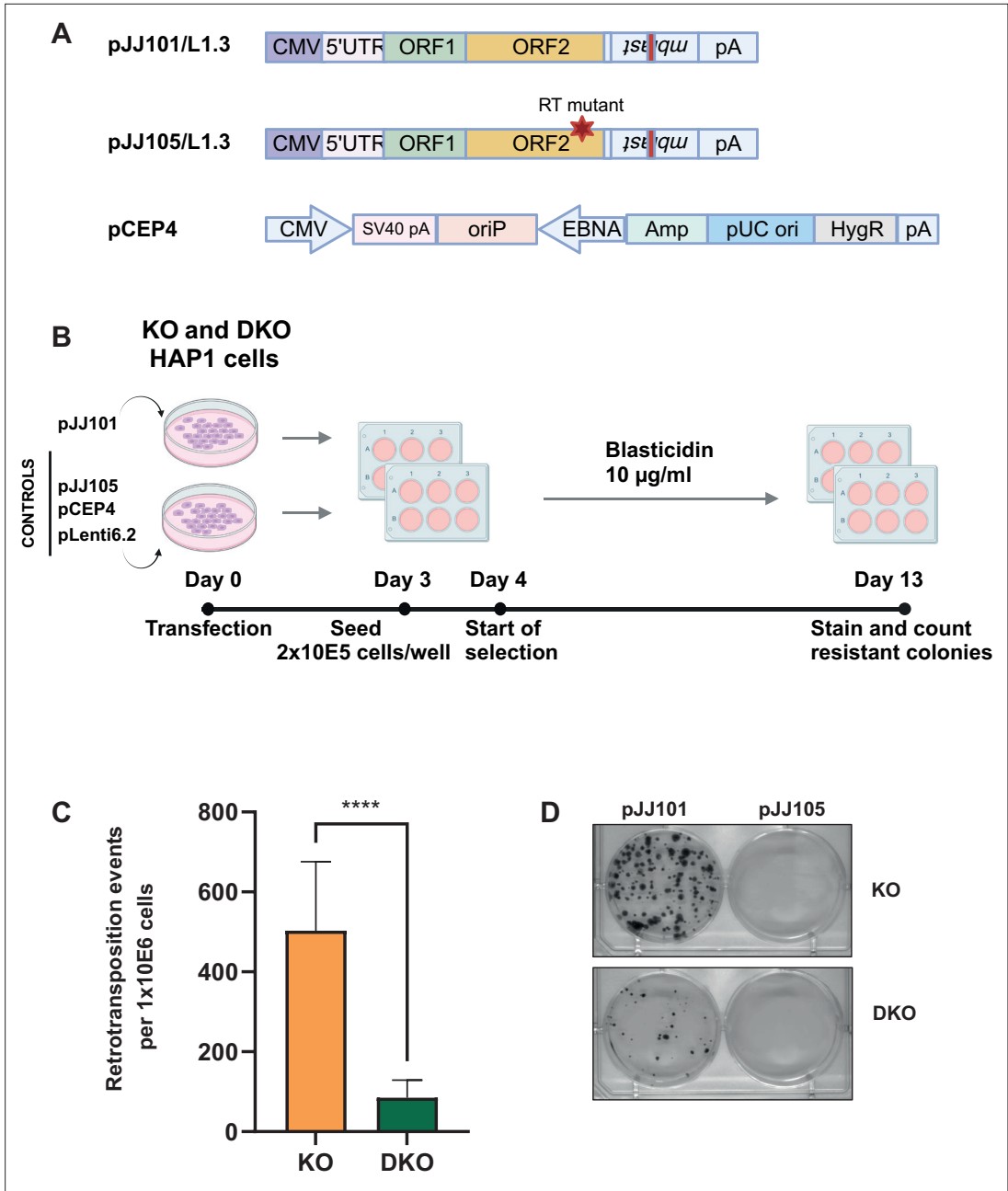

**Figure 4.** L1TD1 promotes L1 retrotransposition. (**A**) Schematic representation of plasmids used for retrotransposition (figure modified from **Kopera et al., 2016** and generated with BioRender.com). The pJJ101/L1.3 construct contains the full-length human L1.3 element with a blasticidin deaminase gene (mblast) inserted in antisense within the 3'UTR. The *mblast* gene is disrupted by an intron and *mblast* expression occurs only when L1 transcript is expressed, reverse transcribed, and inserted into the genome. The pJJ105/L1.3 plasmid contains a mutation in the reverse transcriptase (RT), resulting in defective retrotransposition. The backbone plasmid pCEP4 was used as additional negative control. The blasticidin deaminase gene containing plasmid pLenti6.2 was used as transfection/selection control. (**B**) Workflow of retrotransposition assay. DNMT1 KO and DNMT1/L1TD1 DKO cells were separately transfected with pJJ101 and control plasmids. Equal number of cells were seeded for each condition. Blasticidin selection (10 µg/ml) was started at day 4 and resistant colonies were counted on day 13. This panel was created using BioRender.com. (**C**) Bar graph showing the average number of retrotransposition events per $10^6$ cells seeded in three independent experiments. Blasticidin-resistant colonies in pLenti6.2 transfected cells were used for normalization. Statistical significance was determined using unpaired *t*-test. All data in the figure are shown as a mean of ± SD of three independent experiments, ****p≤0.0001. (**D**) Representative pictures of bromophenol blue stainings of blasticidin-resistant colonies for each genotype and each transfection.

The online version of this article includes the following source data and figure supplement(s) for figure 4:

**Source data 1.** Source data for all plots in *Figure 4* and corresponding details of statistical tests and p-values.

*Figure 4 continued on next page*

*Figure 4 continued*

**Figure supplement 1.** L1TD1 enhances L1 retrotransposition.

**Figure supplement 1—source data 1.** Plots and statistics source data for all plots in *Figure 4—figure supplement 1* and corresponding details of statistical tests and p-values.

described for the nonsense-mediated decay factor UPF1 (*Taylor et al., 2013*). UPF1 knockdown increased the amount of L1 mRNA and proteins but simultaneously reduced the effectiveness of the retrotransposition. A positive effect of L1TD1 on retrotransposition was recently also observed upon L1TD1 overexpression in HeLa cells (*Jin et al., 2024*). The L1TD1 protein shares with its ancestor L1 ORF1p the non-canonical RNA recognition motif and the coiled-coil motif required for the trimerization but has two copies instead of one copy of the C-terminal domain (CTD), a structure with RNA-binding and chaperone function (*Januszyk et al., 2007*; *Martin, 2006*; *McLaughlin et al., 2014*; *Naufer et al., 2019*; *Santos et al., 2015*). We show here that L1TD1 not only binds to L1 transcripts but also associates with L1 ORF1p in an RNA-independent manner. Both features might contribute to the enhanced retrotransposition frequency. These observations are compatible with a model where L1TD1/ORF1p heteromultimers bind to *L1* RNA (*Figure 4—figure supplement 1C*). We hence speculate that the presence of an additional CTD within the L1TD1 protein might thereby enhance the RNA-binding and chaperone function of L1TD1/ORF1p heteromultimers.

To gain insight into the regulatory function of L1TD1 in tumor cells, we performed RIP-seq, transcriptome, and proteome analyses in HAP1 cells. The RIP-seq approach identified, in addition to the known associated *L1TD1* mRNA (*Närvä et al., 2012*), a defined set of transcripts comprising mRNAs, lncRNAs, and TEs including *L1* transcripts. These findings indicate that L1TD1 has not only inherited its function as RNA-binding protein but also its affinity for transposon transcripts. Recently, Jin et al. published a study on the impact of L1TD1 on the translation in hESCs and identified L1TD1-bound RNAs by CLIP-seq (cross-linking immunoprecipitation-high-throughput sequencing) (*Jin et al., 2024*). Interestingly, the majority of L1TD1-associated transcripts in HAP1 cells (69%) identified in our study were also reported as L1TD1 targets in hESCs, suggesting a conserved binding affinity of this domesticated transposon protein across different cell types.

The same study (*Jin et al., 2024*) showed that L1TD1 localizes to high-density RNP condensates and enhances the translation of a subset of mRNAs enriched in the condensates. Our mass spectrometry data showed that L1TD1 ablation leads to a significantly changed proteome. None of the proteins encoded by the mRNAs associated with L1TD1 showed significantly changed steady state levels in the absence of L1TD1, suggesting that this is not a direct effect of L1TD1 association with the respective mRNAs. However, a subset of L1TD1-associated transcripts encode proteins involved in the control of cell division and cell cycle. Thus, it is possible that subtle changes in the expression of this protein that were not detected in our mass spectrometry approach contribute to the antiproliferative effect of L1TD1 depletion (see below).

In accordance with the hESC study (*Jin et al., 2024*), we did not find an overlap between RNAs associated with L1TD1 and their deregulation at the RNA level. This suggests that in HAP1 cells L1TD1 does not have a global effect on RNA stability and translation of its associated RNAs. This is consistent with a recent study on L1 bodies, where RNA processing association of L1 body components did not correlate with RNA stability or translational control of the RNA targets of L1 bodies (*De Luca et al., 2023*). Of note, L1TD1 associated with *SINE* transcripts such as AluY and L1TD1 ablation positively affected AluY transposon expression. This relatively young subfamily of SINE elements is still retrotransposition-competent and their mobilization depends on the activity of LINE proteins (*Roy-Engel, 2012*; *Styles and Brookfield, 2009*). These observations suggest that L1TD1 has not only inherited its function as an RNA-binding protein but also the affinity to *L1* and *SINE* transcripts. L1TD1 association with *AluY* transcripts could potentially be connected to modulation of L1 and SINE-1 transposon activities. It would be interesting to check in future experiments the impact of L1TD1 in Alu retrotransposition reporter assays.

L1 is one of the few protein-coding transposons that is active in humans and L1 overexpression and retrotransposition are hallmarks of cancers (*Kazazian and Moran, 2017*; *Mendez-Dorantes and Burns, 2023*). Similarly to the epigenetic control of the ancestral L1 elements (*Sanchez-Luque et al., 2019*; *Zhou et al., 2020*), L1TD1 expression in HAP1 cells is restricted by DNA methylation and DNA hypomethylation induced by DNMT1 ablation results in robust L1TD1 RNA and protein expression.

DNMT1 ablation in human colorectal carcinoma cells results in mitotic catastrophe and loss of cell viability (*Chen et al., 2007*), whereas HAP1 DNMT1 KO cells show increased DNA damage response and apoptosis but are viable. The less severe phenotype of DNMT1 ablation in HAP1 cells might be due to the compensatory up-regulation of DNMT3A and DNMT3B.

We report here that L1TD1 deletion in DNMT1-deficient cells led to reduced cell viability and increased apoptosis in HAP1 cancer cells. The observation that loss of L1TD1 led to increased apoptosis and DNA damage, but decreased L1 retrotransposition is at first glance unexpected. The positive role of L1TD1 for proliferation might be due to a transposition-independent effects of L1TD1 on the expression of cell cycle regulators as discussed above. Potential deleterious effects of enhanced retrotransposition might be buffered by the upregulation of KRAB domain proteins upon loss of DNMT1. Our findings are in contrast to the negative effect of L1TD1 on cell viability and tumor growth observed in the NSCLC xenograft model (*Altenberger et al., 2017*), but in accordance with the findings that L1TD1 has a positive impact on cell viability in seminoma (*Närvä et al., 2012*) and medullablastoma (*Santos et al., 2015*), suggesting a cancer cell type-specific effect of L1TD1 that might be related to the DNA methylation state of the tumors. Taken together, we present here a novel cellular function for a domesticated transposon protein by showing that L1TD1 associates with L1-RNPs and promotes L1 retrotransposition. This function might have a beneficial effect during early development but could also impact on tumorigenesis.

# Materials and methods

**Key resources table**

| Reagent type (species) or resource | Designation | Source or reference | Identifiers | Additional information |
|---|---|---|---|---|
| Cell line (*Homo sapiens*) | HAP1 | Near-haploid human cell line derived from the male CML cell line KBM-7 | Horizon Discovery C631 RRID:CVCL_Y019 | |
| Cell line (*H. sapiens*) | OV-90 | Human malignant papillary serous carcinoma cell line | ATCC-CRL-3585 RRID:CVCL_3768 | |
| Transfected construct (human) | pJJ101/L1.3 (plasmid) | *Kopera et al., 2016* | Retrotransposition assay | Full-length human L1.3 element |
| Transfected construct (human) | pJJ105/L1.3 (plasmid) | *Kopera et al., 2016* | Retrotransposition assay | Full-length human L1.3 element with mutation in RT |
| Transfected construct (human) | pCEP4 (plasmid) | *Kopera et al., 2016* | Retrotransposition assay | Backbone plasmid |
| Transfected construct (human) | pLenti6.2 | Thermo Fisher | Retrotransposition assay control | Blasticidin deaminase gene |
| Antibody | L1TD1 (polyclonal rabbit) | Sigma-Aldrich | HPA028501 RRID:AB_10599287 | 1:1000 |
| Antibody | L1TD1 (monoclonal mouse) | R&D Systems | MAB8317 RRID:AB_10616625 | 4 ug/IP |
| Antibody | LINE-1 ORF1p (monoclonal mouse) | Sigma-Aldrich | MABC1152 RRID:AB_2941775 | 4 ug/IP 1:1000 WB 1:1000 IF |
| Antibody | LAMIN (polyclonal goat) | Santa Cruz | sc-6216 RRID:AB_648156 | 1:1000 |
| Antibody | Vinculin (monoclonal rabbit) | Cell Signaling | 13901 RRID:AB_2728768 | 1:1000 |
| Antibody | β-actin (monoclonal mouse) | Abcam | ab8226 RRID:AB_306371 | 1:1000 |

*Continued on next page*

*Continued*

| Reagent type (species) or resource | Designation | Source or reference | Identifiers | Additional information |
|---|---|---|---|---|
| Antibody | DNMT1 (H-300) (polyclonal rabbit) | Santa Cruz | sc20701 RRID:AB_2293064 | 1:1000 |
| Antibody | γ-H2AX (monoclonal mouse) | Millipore | JBW301 RRID:AB_2847865 | 1:1000 |
| Antibody | Histone H3 C-term (polyclonal rabbit) | Abcam | Ab1791 RRID:AB_302613 | 1:5000 |
| Antibody | Cleaved caspase 3 (polyclonal rabbit) | Cell Signaling | 9661 RRID:AB_2341188 | 1:50 |
| Antibody | Anti-mouse HRP (polyclonal) | Jackson Laboratories | 115-035-008 RRID:AB_2313585 | 1:10,000 |
| Antibody | Anti-rabbit HRP (polyclonal) | Jackson Laboratories | 211-032-171 RRID:AB_2339149 | 1:10,000 |
| Antibody | Goat anti-Rabbit IgG (H+L) Alexa Fluor Plus 488 | Invitrogen | A32731 RRID:AB_2633280 | 1:500 |
| Antibody | Goat anti-Mouse IgG (H+L) Alexa Fluor Plus 546 | Invitrogen | A11030 RRID:AB_2737024 | 1:500 |
| Sequence-based reagent | h*L1TD1* _f | This paper | PCR primers | CTTACCCTGG TAGCCGACCT |
| Sequence-based reagent | h*L1TD1* _r | This paper | PCR primers | GGCTGGCAAA TTTTCTAAGG |
| Sequence-based reagent | h*ARMC1*_f | This paper | PCR primers | AGCTCTGGAG CGAATTTAAGA |
| Sequence-based reagent | h*ARMC1*_r | This paper | PCR primers | GGCAGACATC CCTGATCCTG |
| Sequence-based reagent | h*YY2*_f | This paper | PCR primers | TCCCGGATAG CATTGAAGAC |
| Sequence-based reagent | h*YY2*_r | This paper | PCR primers | TTGACCTGCA TTTGCTTCTG |
| Sequence-based reagent | h*ORF1p*_f | This paper | PCR primers | AGTGCTTAAAG GAGCTGAT GG |
| Sequence-based reagent | h*ORF1p*_r | This paper | PCR primers | AACTGGAAGAA AGGGTATC AGC |
| Commercial assay or kit | CellTiter-Glo Luminiscent Cell Viability Assay | Promega | G7571 | |
| Commercial assay or kit | Monarch RNA Cleanup Kit | New England Biolabs | T2047L | |
| Commercial assay or kit | Qubit RNA High Sensitivity kit | Thermo Fisher Scientific | Q32852 | |
| Commercial assay or kit | iScript cDNA synthesis Kit | Bio-Rad | 1708891 | |
| Commercial assay or kit | Wizard Genomic DNA isolation kit | Promega | | |
| Commercial assay or kit | EZ DNA Methylation Kit | Zymo Research | D5001 | |
| Software, algorithm | DESeq2 | *Love et al., 2014* | RRID:SCR_015687 | |
| Software, algorithm | TEtranscript | *Jin et al., 2015* | RRID:SCR_023208 | |

*Continued on next page*

*Continued*

| Reagent type (species) or resource | Designation | Source or reference | Identifiers | Additional information |
|---|---|---|---|---|
| Software, algorithm | Fiji ImageJ | *Schindelin et al., 2012* | RRID:SCR_002285 | |
| Software, algorithm | FlowJo | BD Bioscience | 10.6.1 RRID:SCR_008520 | |

## Cell lines and cell culture

HAP1 is a near-haploid human cell line derived from the KBM-7 chronic myelogenous leukemia (CML) cell line (*Andersson et al., 1987*; *Carette et al., 2011*). Mycoplasma-free HAP1 cells were provided by Horizon Genomics (now Horizon Discovery). HAP1 cells were cultured in Iscove's Modified Dulbecco Medium (IMDM, Sigma-Aldrich, I3390), supplemented with 10% fetal bovine serum (Sigma-Aldrich, F7524), and 1% penicillin/streptomycin (Sigma-Aldrich, P4333). The human malignant papillary serous carcinoma cell line OV-90 (*Gralewska et al., 2021*) was cultured in MCDB 105 (Sigma-Aldrich, 117) and Medium 199 (Sigma-Aldrich, M4530) in a 1:1 ratio, supplemented with 15% fetal bovine serum (Sigma-Aldrich, F7524), and 1% penicillin/streptomycin (Sigma-Aldrich, P4333). The cells were kept in a humidified incubator at 37°C and 5% $CO_2$. Cell lines were regularly tested for mycoplasma.

## Gene editing

DNMT1 KO HAP1 cells harboring a 20 bp deletion in the exon 4 of the *DNMT1* gene were generated using CRISPR/Cas9 gene editing. DNMT1/L1TD1 DKO HAP1 cells were generated by CRISPR/Cas9 gene editing, resulting in a 13 bp deletion in exon 4 of the L1TD1 gene. CRISPR/Cas9 gene editing was performed by Horizon Genomics (now Horizon Discovery).

## Viability and apoptosis assay

Both assays were performed as described previously (*Hess et al., 2022*). For the viability assay, $2 \times 10^3$ cells of each cell line were seeded in triplicates in solid white 96-well plates and allowed to attach to the plates overnight. Cell viability was measured using CellTiter-Glo Luminiscent Cell Viability Assay (Promega, G7571) and Glomax Discover plate reader (Promega). The measurement was performed every 24 hours, starting from the day after seeding (24, 48, 72 hours).

For the apoptosis assay, $5 \times 10^6$ cells were collected 48 hours after seeding, fixed for 20 min 2% paraformaldehyde (Merck Life Science), and for 30 min in 75% ethanol. Cells were permeabilized with 0.1% Triton X-100 (Merck Life Science) for 10 min, blocked in 10% donkey serum (Merck Life Science) for 30 min, and then incubated with 1:50 diluted anti-cleaved caspase 3 antibody (Cell Signaling, 9661) in phosphate-buffered saline (PBS) for 1 hour. Cells were washed and incubated with 1:400 Alexa 488 anti-rabbit secondary antibody (Jackson ImmunoResearch, 751-545-152). Samples were acquired using FACSCelesta flow cytometer (BD Bioscience), and the data were analyzed using FlowJo software 10.6.1 (BD Bioscience).

## Immunoprecipitation

HAP1 cells pellets harvested from 15 cm culture plates were lysed in Hunt Buffer (20 mM Tris/HCl pH 8.0, 100 mM NaCl, 1 mM EDTA, 0.5% NP-40), supplemented with Complete Protease inhibitor cocktail (Roche, 11697498001), Complete Phosphatase inhibitor cocktail (Roche, 11836145001), 10 mM sodium fluoride, 10 mM β-glycerophosphate, 0.1 mM sodium molybdate, and 0.1 mM PMSF. The lysis was achieved by repeating freeze–thaw cycles three times followed by centrifugation 12,000 × *g* for 15 min at 4°C. Cell lysate was collected and the protein concentration was measured using Bradford assay. 40 µl of Dynabeads Protein G beads (Thermo Fisher Scientific, 10004D) was blocked with 10% bovine serum albumin (BSA) in a rotor at 4°C for 1 hour. 1 mg of cell lysate was incubated with blocked Protein G beads and monoclonal L1 ORF1p antibody (EMD Millipore, MABC1152) at 12 rpm at 4°C overnight. The next day, the beads were washed three times with Hunt buffer.

## Western blot analysis

Immunoprecipitated complex was eluted from the beads with 1× SDS loading dye followed by heat-incubation at 95℃ for 5 min for detection of proteins. For input samples, 20 µg whole cell

lysate was similarly denatured as IP samples. Proteins were separated in SDS-polyacrylamide gel and transferred onto a nitrocellulose membrane (Amersham Protran, GE10600001, Sigma-Aldrich) by wet transfer method. The membrane was blocked with the blocking solution (1× PBS, 1% polyvinylpyrrolidone, 3% non-fat dried milk, 0.1% Tween-20, 0.01% sodium azide, pH 7.4) and incubated with the corresponding antibodies listed in *Supplementary file 8*. To detect the protein of interest, ECL western blotting detection reagents were used with a FUSION FX chemiluminescence imaging system.

## Mass spectrometry analysis

Proteomes of DNMT1 KO and DNMT1/L1TD1 DKO cells were analyzed by quantitative TMT (Tandem Mass Tag) multiplex mass spectrometry analysis. Proteins were precipitated from cell extracts with acetone. After reduction and alkylation of Cys under denaturing conditions, proteins were digested with LysC/trypsin overnight. Peptides were cleaned-up using Oasis MCX (Waters) and labeled with TMT6plex (Thermo Fisher) according to the manufacturer's protocol. Equal amounts of each channel were mixed, desalted, and fractionated by neutral pH reversed-phase chromatography. Equal interval fractions were pooled and analyzed by LC-MS3 on an Ultimate 3000 RSLCnano LC coupled to an Orbitrap Eclipse Tribrid mass spectrometer via a FAIMS Pro ion mobility interface (all Thermo Fisher). Data was analyzed with MaxQuant 1.6.7.0. (*Tyanova et al., 2016*). Differentially enriched proteins were determined with LIMMA (*Ritchie et al., 2015*). Workflow and analysis are described in detail in *Supplementary file 4*.

## RNA immunoprecipitation and RIP-qRT-PCR

The native RIP protocol was followed as previously described in *Yang et al., 2017*. In brief, cells were resuspended in 1 ml polysomal lysis buffer (100 mM KCl, 5 mM $MgCl_2$, 10 mM HEPES [pH 7.0], 0.5% NP-40, 1 mM DTT), supplemented with Protector RNase inhibitor (Sigma-Aldrich, 3335399001), and protease inhibitor cocktail (Roche, 11697498001). Cell lysis was facilitated with 27G needle and syringe at least seven times. The lysates were centrifuged at 12,000 × *g* for 15 min at 4°C. 1% input was taken from the cell lysate for RIP-seq experiment. The supernatant was cleared via pre-incubation with Dynabeads Protein G beads (Thermo Fisher Scientific, 10004D) rotating for 1 hour at 4°C. For immunoprecipitation, 1 mg of pre-cleaned cell lysate was incubated with 4 µg mouse monoclonal L1TD1 antibody (R&D Systems, MAB8317) at 4°C at 20 rpm on the rotor overnight. 40 µl Dynabeads Protein G beads per IP was blocked with 10% BSA for 1 hour at 4°C. Blocked beads were added to the lysate and the antibody mixture was rotating for 1 hour at 4°C. Next, the beads were pelleted and washed three times with polysomal lysis buffer. 20% of the RIP was used for western blot analysis to confirm the presence of immunoprecipitated L1TD1. 1% of the RIP was saved as input. All experiments were performed in biological triplicates. Validation of the RIP experiments by RIP-qRT-PCR was performed using the Sigma-Aldrich RIP-qRT-PCR: Data Analysis Calculation Shell. Fold enrichment of transcripts in the IP samples was calculated as ratio of qRT-PCR values of KO cells representing the specific association of the transcript to L1TD1 relative the qRT-PCR values of DKO cells representing unspecific binding in the absence of L1TD1 (arbitrarily set to 1) and normalized to the qRT-values of input samples in the indicated cells. For L1, primers specific for the L1.2 subfamily were used.

## RNA isolation

To extract total RNA, immunoprecipitated RNA and 1% input RNA Monarch RNA Cleanup Kit (New England Biolabs, T2047L) was used according to the manufacturer's instructions. In the case of RIP samples, the beads were first treated with DNase I (New England Biolabs, M0303S) at 37°C for 15 min and 1 ml TRIzol (Thermo Fisher Scientific, 15596018) was directly added on. Upon addition of 150 µl chloroform, the mixture was vortexed vigorously for 15 s and centrifuged for at 12,000 × *g* for 15 min at 4°C. Aqueous phase was transferred to a new Eppendorf tube and mixed with 1 volume of EtOH (>95%). The mixture was loaded on the RNA columns and spun for 1 min. The columns were washed two times with the washing buffer (supplemented with the kit) and eluted in 20 µl nuclease-free water. RNA concentration was measured using the Qubit RNA High Sensitivity kit (Thermo Fisher Scientific, Q32852).

## cDNA library preparation

The cDNA libraries for the RIP-seq experiment were prepared as described in Smart-seq3 protocol by *Hagemann-Jensen et al., 2020* (*Hagemann-Jensen et al., 2020*).

## cDNA synthesis and qRT-PCR

1 µg of total RNA was reverse transcribed using iScript cDNA synthesis Kit (Bio-Rad, 1708891). Reverse transcription reaction was performed as follows: 25°C for 5 min, 46°C for 30 min, 95°C for 5 min. The resulting cDNA was diluted to 1:10 with nuclease-free water. 5 µl of the cDNA was used for SYBR Green qPCR Master mix (Bio-Rad 1725275) together with the primers used in this study, which are listed in *Supplementary file 7*.

## MethyLight assay

Genomic DNA was extracted from DNMT1 KO, DNMT1/L1TD1 DKO, and WT HAP1 cells using the Wizard Genomic DNA isolation kit (Promega) following the manufacturer's protocol. Next, 1 µg of genomic DNA per sample was subjected to bisulfite conversion using the EZ DNA Methylation Kit (Zymo Research D5001) following the manufacturer's instructions. Bisulfite-treated DNA was eluted in ddH$_2$O to a final concentration of 10 ng/µl and stored at –20°C until further use. The MethyLight method was performed as previously described by *Campan et al., 2009*, using primers for *L1TD1* and *L1; ALU* was used as a reference (*Weisenberger et al., 2005*). The primer sequences are provided in *Supplementary file 7*. The MethyLight reactions were done in technical triplicates. Each reaction contained 7.5 µl of 2X TaqMan Universal PCR Master Mix (Thermo Fisher 4324018), 300 nM of each primer and 100 nM of probe. For the *L1TD1* reactions 50 ng and for the *L1* reactions 20 ng of bisulfite-converted DNA were used in a total volume of 15 µl. The reactions were performed on a Bio-Rad CFX96 thermocycler with an initial incubation at 95°C for 10 min, followed by 50 cycles of 95°C for 15 s and 60°C for 1 min. Each run included the individual samples, a serial dilution of fully methylated DNA, a non-methylated DNA control (Human Methylated & Non-Methylated [WGA] DNA Set D5013 Zymo Research), and a non-template control. The percentage of methylation reference (PMR) of each individual sample was calculated using the following formula: 100 × [(GENE-X mean value) sample/(ALU mean value) sample]/[(GENE-X mean value) 100% methylated/(ALU mean value)100% methylated].

## RNA-seq and RIP-seq analyses

RNA sequencing data were analyzed using the RNA-Seq pipeline of IMP/IMBA Bioinfo core facility (ii-bioinfo@imp.ac.at). The pipeline is based on the nf-core/rnaseq pipeline (https://doi.org/10.5281/zenodo.1400710) and is built with nextflow (*Ewels et al., 2020*).The RNA-seq analysis was performed as described in the following: Adapters were clipped with trimgalore (*Martin, 2011*). Abundant sequence fractions (rRNA) were removed using (bowtie2) (*Langmead and Salzberg, 2012*). Cleaned raw reads were mapped against the reference genome (hg38,Homo_sapiens.GRCh38.107.gtf) with STAR (reverse_stranded, https://github.com/alexdobin/STAR, copy archived at *Doblin, 2024*; *Dobin et al., 2013*). Mapped reads were assigned to corresponding genes using featureCounts (*Liao et al., 2014*). Abundances were estimated through Kallisto (*Bray et al., 2016*) and Salmon (*Patro et al., 2017*). Analysis of differentially expressed genes was performed using DESeq2 (*Love et al., 2014*). TEtranscript software (*Jin et al., 2015*) was used to identify differentially expressed TE-derived sequences. In the case of RIP-seq, differential enrichment of transcripts relative to input as well as to the negative control (HAP1 DNMT1/L1TD1 DKO) was calculated using DEseq2.

## L1 retrotransposition assay

Transfection of HAP1 cells with pJJ101, pJJ105, EGFP, pLenti6.2, pCEP4 was performed using polyethyleneimine (PEI)/NaCl solution. Briefly, 0.2 × 10$^6$ HAP1 cells/well were seeded into 6-well plates. Twenty-four hours later and at 70% confluence, medium was replaced by fresh IMDM without penicillin/streptavidin followed by transfection. For each transfection, 100 µl of 150 mM NaCl, 7 µl of PEI were mixed with DNA/NaCl solution (100 µl of 150 mM NaCl, 4 µg of plasmid DNA), incubated at room temperature for 30 min, followed by adding the mixture in a drop-wise fashion onto cells. Eighteen hours post-transfection, the media was replaced with complete IMDM including 1% penicillin/

streptavidin. Transfection efficiency was monitored 48 hours after transfection using fluorescent microscopy.

The retrotransposition assay was performed as described in *Kopera et al., 2016* with the following modification. To measure retrotransposition efficiency, $2×10^5$ cells per well were seeded in 6-well culture plates and cultured at 37°C overnight. On day 0.4 µg of pJJ101/L1.3, pJJ105/L1.3, pLenti6.2, pCEP4 vectors (described in *Kopera et al., 2016*) were transfected separately using PEI/NaCl in DNMT1 KO and DNMT1/L1TD1 DKO cells. To correct for potential differences in transfection efficiency and susceptibility to blasticidin in DNMT1 KO and DNMT1/L1TD1 DKO cells, we transfected DNMT1 KO and DNMT1/L1TD1 DKO cells in parallel with the blasticidin deaminase containing vector pLenti6.2 as control. The following day (d1) the media was replaced with IMDM medium containing 1% penicillin/streptavidin. On day 3, $2×10^5$ cells per well seeded into 6-well culture dishes for each genotype and condition. Blasticidin treatment 10 µg/ml was started on day 4 and the cells were cultured 37°C for 9 days without medium change. Blasticidin-resistant colonies were fixed on day 13 with 4% methanol-free formaldehyde (Thermo Fisher Scientific, 28908) at room temperature for 20 min and stained with 0.1% bromophenol blue (w/v) at room temperature for 1 hour. The pictures of the wells containing the colonies were taken using the FX chemiluminescence imaging system. The pictures were further processed in Fiji Software following the Analyze Particles function (*Schindelin et al., 2012*). The colony counts were obtained from three technical replicates per transfection. Mean colony counts were calculated and adjusted retrotransposition mean was calculated by adjusting for blasticidin-resistant colonies in blasticidin vector (pLenti6.2) transfected control cells.

## Indirect immunofluorescence staining

For immunostaining, the protocol from *Sharma et al., 2016* was followed with minor modifications. Briefly, $75 × 10^3$ HAP1 cells per well were plated on cover slips in 12-well plates. Next day, the cells were washed with 1× PBS and fixed with 4% methanol-free formaldehyde at room temperature for 10 min. The cells were washed with 1× PBS/glycine pH 7.4 and permeabilized with 0.1% Triton-X in 1× PBS/glycine for 3 min. The samples were blocked with 1× PBS/glycine/1% BSA at room temperature for 1 hour. After blocking, the cells were stained primary antibodies in blocking solution at 4°C overnight. On the following day, the cells were washed three times with 1× PBS/glycine/1% BSA for 5 min and stained with secondary antibody in blocking solution for 2 hours in the dark. After a wash with PBS/glycine pH 7.4, cells were incubated with DAPI (Sigma-Aldrich, D9542, 1:10,000) in PBS/glycine at room temperature for 10 min and washed three times with 1× PBS/glycine pH 7.4. The slides were mounted with ProLong Gold Mountant (Thermo Fisher Scientific, P36930) and dried. The staining was analyzed using an Olympus Confocal Microscope.

## Acknowledgements

We would like to thank Wolfgang Sommergruber for the screen of human tumor cells for L1TD1 expression, Jernej Ule for help with the analysis of RNPs, and John Moran for plasmids for the retrotransposition assay. We also want to thank Luisa Seufert, Stephanie Schneider, Christina Maria Schuh, and Marlene Müller for help with the characterization of transgenic HAP1 cells; Brigitte Gundacker, Urska Janjos, and Milena Mijovic for professional technical support; and Dorothea Anrather and Claudia Stocsits for help with data analysis. We are also grateful to Wolfgang Miller, Heinz Fischer, and Matthias Schaefer for numerous helpful discussions and Justin Trowbridge for helpful comments on the manuscript. GE and TM by the Austrian Science Fund (FWF, doc.funds grant DOC59), CS was supported by the Austrian Science Fund (P34998, DOC32) and the Austrian Research Promotion Agency (FFG): Bridge Early Phase project 5722451. GK was a PhD student of the doc.funds program (DOC32) supported by the Austrian Science Fund (FWF). TPC was supported by a Student Fellowship from the Ernst Mach Grant – ASEA-UNINET through the Austrian Academic Exchange (OeAD). For the purpose of open access, the authors have applied a CC BY public copyright license to any Author Accepted Manuscript version arising from this submission.

## Additional information

### Funding

| Funder | Grant reference number | Author |
|---|---|---|
| Austrian Science Fund | 10.55776/P34998 | Christian Seiser |
| Austrian Science Fund | DOC32 | Gülnihal Kavaklioglu Christian Seiser |
| Austrian Science Fund | DOC59 | Theresia Mair Gerda Egger |
| Austrian Research Promotion Agency | Bridge Early Phase project 5722451 | Christian Seiser |
| Austrian-South-East Asian Academic University Network | Ernst Mach Grant | Trinh Phan-Canh |

The funders had no role in study design, data collection and interpretation, or the decision to submit the work for publication.

### Author contributions

Gülnihal Kavaklioglu, Conceptualization, Data curation, Validation, Investigation, Visualization, Writing – original draft; Alexandra Podhornik, Validation, Investigation, Visualization; Terezia Vcelkova, Resources, Investigation, Methodology; Jelena Marjanovic, Theresia Mair, Claudia Miccolo, Mirko Doni, Investigation, Visualization, Methodology; Mirjam A Beck, Conceptualization, Data curation, Validation, Investigation; Trinh Phan-Canh, Formal analysis, Investigation, Visualization; Aleksej Drino, Conceptualization, Methodology; Gerda Egger, Susanna Chiocca, Conceptualization, Validation, Methodology; Miha Modic, Conceptualization, Supervision, Methodology; Christian Seiser, Conceptualization, Resources, Data curation, Funding acquisition, Writing – original draft, Project administration, Writing – review and editing

### Author ORCIDs

Trinh Phan-Canh https://orcid.org/0000-0002-6399-0959
Gerda Egger https://orcid.org/0000-0003-2489-155X
Christian Seiser https://orcid.org/0000-0002-7046-9352

Reviewer #1 (Public review): https://doi.org/10.7554/eLife.96850.4.sa1
Reviewer #2 (Public review): https://doi.org/10.7554/eLife.96850.4.sa2
Author response https://doi.org/10.7554/eLife.96850.4.sa3

## Additional files

### Supplementary files

Supplementary file 1. DEG list (upregulated and downregulated) for HAP1 DNMT1 KO versus HAP1 WT cells.

Supplementary file 2. DESeq2 analysis of L1TD1 RIP-seq data from HAP1 DNMT1 KO and HAP1 DNMT1 L1TD1 DKO cells.

Supplementary file 3. TETranscript analysis of L1TD1 RIP-seq data from HAP1 DNMT1 KO and HAP1 DNMT1 L1TD1 DKO cells.

Supplementary file 4. Mass spectrometry analysis of deregulated proteins in HAP1 DNMT1 KO and HAP1 DNMT1 L1TD1 DKO cells.

Supplementary file 5. DESeq2 transcriptome analysis of HAP1 DNMT1 L1TD1 DKO versus HAP1 DNMT1 KO cells.

Supplementary file 6. t transcriptome analysis of HAP1 DNMT1 L1TD1 DKO versus HAP1 DNMT1 KO cells.

Supplementary file 7. Sequences of primers used in this study.

Supplementary file 8. Antibodies used in this study.

MDAR checklist

## Data availability

RNA-seq and RIP-seq data were submitted to GEO under accession numbers GSE169614, GSE254459 and GSE254460. The mass spectrometry proteomics data have been deposited to the ProteomeXchange Consortium via the PRIDE (*Perez-Riverol et al., 2022*) partner repository with the dataset identifier PXD047402.

The following datasets were generated:

| Author(s) | Year | Dataset title | Dataset URL | Database and Identifier |
|---|---|---|---|---|
| Beck MA, Gülnihal K, Tamir IM, Seiser C | 2024 | RNA sequencing of control, DNMT1 knockout and DNMT1/L1TD1 double knockout haploid cells (HAP1) | https://www.ncbi.nlm.nih.gov/geo/query/acc.cgi?acc=GSE169614 | NCBI Gene Expression Omnibus, GSE169614 |
| Seiser C | 2025 | Identification of L1TD1-associated transcripts in DNMT1 KO HAP1 cells | https://www.ncbi.nlm.nih.gov/geo/query/acc.cgi?acc=GSE254459 | NCBI Gene Expression Omnibus, GSE254459 |
| Seiser C | 2025 | Expression profile of DNMT1 KO and DNMT1 L1TD1 HAP1 cells by high throughput sequencing | https://www.ncbi.nlm.nih.gov/geo/query/acc.cgi?acc=GSE254460 | NCBI Gene Expression Omnibus, GSE254460 |
| Anrather D | 2025 | Potential effect of L1TD1 on proteomics in HAP1 cells | https://www.ebi.ac.uk/pride/archive/projects/PXD047402 | PRIDE, PXD047402 |

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
