## [Editor Report · eLife Assessment]

This **important** article reports functional interactions between L1TD1, an RNA-binding protein (RBP), and its ancestral LINE-1 retrotransposon, which is not modulated at the translational level. The evidence for the association between L1TD1 and LINE-1 ORF1p is **solid**. The work implies that the transposon-derived RNA-binding protein in the human genome can interact with the ancestral transposable element from which this protein was initially derived. This work spurs interesting questions for cancer types, where LINE1 and L1TD1 are aberrantly expressed.

---

## [Referee Report · Reviewer #1 (Public review)]

Summary:

In their manuscript entitled 'The domesticated transposon protein L1TD1 associates with its ancestor L1 ORF1p to promote LINE-1 retrotransposition', Kavaklıoğlu and colleagues delve into the role of L1TD1, an RNA binding protein (RBP) derived from a LINE1 transposon. L1TD1 proves crucial for maintaining pluripotency in embryonic stem cells and is linked to cancer progression in germ cell tumors, yet its precise molecular function remains elusive. Here, the authors uncover an intriguing interaction between L1TD1 and its ancestral LINE-1 retrotransposon.

The authors delete the DNA methyltransferase DNMT1 in a haploid human cell line (HAP1), inducing widespread DNA hypo-methylation. This hypomethylation prompts abnormal expression of L1TD1. To scrutinize L1TD1's function in a DNMT1 knock-out setting, the authors create DNMT1/L1TD1 double knock-out cell lines (DKO). Curiously, while the loss of global DNA methylation doesn't impede proliferation, additional depletion of L1TD1 leads to DNA damage and apoptosis.

To unravel the molecular mechanism underpinning L1TD1's protective role in the absence of DNA methylation, the authors dissect L1TD1 complexes in terms of protein and RNA composition. They unveil an association with the LINE-1 transposon protein L1-ORF1 and LINE-1 transcripts, among others.

Surprisingly, the authors note fewer LINE-1 retro-transposition events in DKO cells compared to DNMT1 KO alone.

Strengths:

The authors present compelling data suggesting the interplay of a transposon-derived human RNA binding protein with its ancestral transposable element. Their findings spur interesting questions for cancer types, where LINE1 and L1TD1 are aberrantly expressed.

Weaknesses:

The finding that L1TD1/DNMT1 DKO cells exhibit increased apoptosis and DNA damage but decreased L1 retro-transposition is unexpected. Considering the DNA damage associated with retro-transposition and the DNA damage and apoptosis observed in L1TD1/DNMT1 DKO cells, one would anticipate the opposite outcome. Could it be that the observation of fewer transposition-positive colonies stems from the demise of the most transposition-positive colonies? Future studies are bound to further explore this intriguing phenomenon.

---

## [Referee Report · Reviewer #2 (Public review)]

In this study, Kavaklıoğlu et al. investigated and presented evidence for a role for domesticated transposon protein L1TD1 in enabling its ancestral relative, L1 ORF1p, to retrotranspose in HAP1 human tumor cells. The authors provided insight into the molecular function of L1TD1 and shed some clarifying light on previous studies that showed somewhat contradictory outcomes surrounding L1TD1 expression. Here, L1TD1 expression was correlated with L1 activation in a hypomethylation dependent manner, due to DNMT1 deletion in HAP1 cell line. The authors then identified L1TD1 associated RNAs using RIP-Seq, which display a disconnect between transcript and protein abundance (via Tandem Mass Tag multiplex mass spectrometry analysis). The one exception was for L1TD1 itself, is consistent with a model in which the RNA transcripts associated with L1TD1 are not directly regulated at the translation level. Instead, the authors found L1TD1 protein associated with L1-RNPs and this interaction is associated with increased L1 retrotransposition, at least in the contexts of HAP1 cells. Overall, these results support a model in which L1TD1 is restrained by DNA methylation, but in the absence of this repressive mark, L1TD1 is expression, and collaborates with L1 ORF1p (either directly or through interaction with L1 RNA, which remains unclear based on current results), leads to enhances L1 retrotransposition. These results establish feasibility of this relationship existing in vivo in either development or disease, or both.

Comments on revised version:

Thank you for this revised manuscript and for addressing our concerns and suggestions. These improvements have significantly enhanced the quality and reliability of the results presented and have addressed all our questions.

---

## [Author Response]

The following is the authors’ response to the previous reviews.

**Public Reviews:**

**Reviewer #1 (Public review):**
Summary:In their manuscript entitled 'The domesticated transposon protein L1TD1 associates with its ancestor L1 ORF1p to promote LINE-1 retrotransposition', Kavaklıoğlu and colleagues delve into the role of L1TD1, an RNA binding protein (RBP) derived from a LINE1 transposon. L1TD1 proves crucial for maintaining pluripotency in embryonic stem cells and is linked to cancer progression in germ cell tumors, yet its precise molecular function remains elusive. Here, the authors uncover an intriguing interaction between L1TD1 and its ancestral LINE-1 retrotransposon.The authors delete the DNA methyltransferase DNMT1 in a haploid human cell line (HAP1), inducing widespread DNA hypo-methylation. This hypomethylation prompts abnormal expression of L1TD1. To scrutinize L1TD1's function in a DNMT1 knock-out setting, the authors create DNMT1/L1TD1 double knock-out cell lines (DKO). Curiously, while the loss of global DNA methylation doesn't impede proliferation, additional depletion of L1TD1 leads to DNA damage and apoptosis.To unravel the molecular mechanism underpinning L1TD1's protective role in the absence of DNA methylation, the authors dissect L1TD1 complexes in terms of protein and RNA composition. They unveil an association with the LINE-1 transposon protein L1-ORF1 and LINE-1 transcripts, among others.Surprisingly, the authors note fewer LINE-1 retro-transposition events in DKO cells compared to DNMT1 KO alone.Strengths:The authors present compelling data suggesting the interplay of a transposon-derived human RNA binding protein with its ancestral transposable element. Their findings spur interesting questions for cancer types, where LINE1 and L1TD1 are aberrantly expressed.Weaknesses:Suggestions for refinement:The initial experiment, inducing global hypo-methylation by eliminating DNMT1 in HAP1 cells, is intriguing and warrants more detailed description. How many genes experience misregulation or aberrant expression? What phenotypic changes occur in these cells? Why did the authors focus on L1TD1? Providing some of this data would be helpful to understand the rationale behind the thorough analysis of L1TD1.The finding that L1TD1/DNMT1 DKO cells exhibit increased apoptosis and DNA damage but decreased L1 retro-transposition is unexpected. Considering the DNA damage associated with retro-transposition and the DNA damage and apoptosis observed in L1TD1/DNMT1 DKO cells, one would anticipate the opposite outcome. Could it be that the observation of fewer transposition-positive colonies stems from the demise of the most transpositionpositive colonies? Further exploration of this phenomenon would be intriguing.
**Reviewer #2 (Public review):**
In this study, Kavaklıoğlu et al. investigated and presented evidence for a role for domesticated transposon protein L1TD1 in enabling its ancestral relative, L1 ORF1p, to retrotranspose in HAP1 human tumor cells. The authors provided insight into the molecular function of L1TD1 and shed some clarifying light on previous studies that showed somewhat contradictory outcomes surrounding L1TD1 expression. Here, L1TD1 expression was correlated with L1 activation in a hypomethylation dependent manner, due to DNMT1 deletion in HAP1 cell line. The authors then identified L1TD1 associated RNAs using RIPSeq, which display a disconnect between transcript and protein abundance (via Tandem Mass Tag multiplex mass spectrometry analysis). The one exception was for L1TD1 itself, is consistent with a model in which the RNA transcripts associated with L1TD1 are not directly regulated at the translation level. Instead, the authors found L1TD1 protein associated with L1-RNPs and this interaction is associated with increased L1 retrotransposition, at least in the contexts of HAP1 cells. Overall, these results support a model in which L1TD1 is restrained by DNA methylation, but in the absence of this repressive mark, L1TD1 is expression, and collaborates with L1 ORF1p (either directly or through interaction with L1 RNA, which remains unclear based on current results), leads to enhances L1 retrotransposition. These results establish feasibility of this relationship existing in vivo in either development or disease, or both.Comments on revised version:In general, the authors did an acceptable job addressing the major concerns throughout the manuscript. This revision is much clearer and has improved in terms of logical progression.
**Recommendations for the authors:**

**Reviewer #1 (Recommendations for the authors):**
The authors have addressed all my questions in the revised version of the manuscript.
**Reviewer #2 (Recommendations for the authors):**
Revised comments:A few points we'd like to see addressed are our comments about the model (Figure S7C), as this is important for the readership to understand this complex finding. Please try to apply some quantification, if possible (question 8). Please do your best to tone down the direct relationship of these findings to embryology (question 11). Based on both reviewer comments, we believe addressing reviewer #1s "Suggestions for refinement" (2 points), would help us change our view of solid to convincing.Responses to changes:Major(1) The study only used one knockout (KO) cell line generated by CRISPR/Cas9.Considering the possibility of an off-target effect, I suggest the authors attempt one or both of these suggestions.A) Generate or acquire a similar DMNT1 deletion that uses distinct sgRNAs, so that the likelihood of off-targets is negligible. A few simple experiments such as qRT-PCR would be sufficient to suggest the same phenotype.B) Confirm the DNMT1 depletion also by siRNA/ASO KD to phenocopy the KO effect.(2) In addition to the strategies to demonstrate reproducibility, a rescue experiment restoring DNMT1 to the KO or KD cells would be more convincing. (Partial rescue would suffice in this case, as exact endogenous expression levels may be hard to replicate).We have undertook several approaches to study the effect of DNMT1 loss or inactivation: As described above, we have generated a conditional KO mouse with ablation of DNMT1 in the epidermis. DNMT1-deficient keratinocytes isolated from these mice show a significant increase in L1TD1 expression. In addition, treatment of primary human keratinocytes and two squamous cell carcinoma cell lines with the DNMT inhibitor aza-deoxycytidine led to upregulation of L1TD1 expression. Thus, the derepression of L1TD1 upon loss of DNMT1 expression or activity is not a clonal effect.Also, the spectrum of RNAs identified in RIP experiments as L1TD1-associated transcripts in HAP1 DNMT1 KO cells showed a strong overlap with the RNAs isolated by a related yet different method in human embryonic stem cells. When it comes to the effect of L1TD1 on L1-1 retrotranspostion, a recent study has reported a similar effect of L1TD1 upon overexpression in HeLa cells [4].All of these points together help to convince us that our findings with HAP1 DNMT KO are in agreement with results obtained in various other cell systems and are therefore not due to off-target effects. With that in mind, we would pursue the suggestion of Reviewer 1 to analyze the effects of DNA hypomethylation upon DNMT1 ablation.

Thank you for addressing this concern. The reference to Beck 2021 and the additional cells lines (R2: keratinocytes and R3: squamous cell carcinoma) provides sufficient evidence that this result is unlikely to be a result of clonal expansion or off targets.

Question: Was the human ES Cell RIP Experiment shown here? What is the overlap?

We refer to the recently published study by Jin et al. (PMID: 38165001). As stated in the Discussion, the majority of L1TD1-associated transcripts in HAP1 cells (69%) identified in our study were also reported as L1TD1 targets in hESCs suggesting a conserved binding affinity of this domesticated transposon protein across different cell types.

(3) As stated in the introduction, L1TD1 and ORF1p share "sequence resemblance" (Martin 2006). Is the L1TD1 antibody specific or do we see L1 ORF1p if Fig 1C were uncropped?(6) Is it possible the L1TD1 antibody binds L1 ORF1p? This could make Figure 2D somewhat difficult to interpret. Some validation of the specificity of the L1TD1 antibody would remove this concern (see minor concern below).This is a relevant question. We are convinced that the L1TD1 antibody does not crossreact with L1 ORF1p for the following reasons: Firstly, the antibody does not recognize L1 ORF1p (40 kDa) in the uncropped Western blot for Figure 1C (Figure R4A). Secondly, the L1TD1 antibody gives only background signals in DKO cells in the indirect immunofluorescence experiment shown in Figure 1E of the manuscript.Thirdly, the immunogene sequence of L1TD1 that determines the specificity of the antibody was checked in the antibody data sheet from Sigma Aldrich. The corresponding epitope is not present in the L1 ORF1p sequence.Finally, we have shown that the ORF1p antibody does not cross-react with L1TD1 (Figure R4B).Response: Thank you for sharing these images. These full images relieve concerns about specificity. The increase of ORF1P in R4B and Main figure 3C is interesting and pointed out in the manuscript. Not for the purposes of this review, but the observation of reduced transposition despite increased ORF1P could be an interesting follow up to this study (combined with the similar UPF1 result could indicate a complex of some kind).(4) In abstract (P2), the authors mentioned that L1TD1 works as an RNA chaperone, but in the result section (P13), they showed that L1TD1 associates with L1 ORF1p in an RNA independent manner. Those conclusions appear contradictory. Clarification or revision is required.Our findings that both proteins bind L1 RNA, and that L1TD1 interacts with ORF1p are compatible with a scenario where L1TD1/ORF1p heteromultimers bind to L1 RNA. The additional presence of L1TD1 might thereby enhance the RNA chaperone function of ORF1p. This model is visualized now in Suppl. Figure S7C.Response: Thank you for the model. To further clarify, do you mean that L1TD1 can bind L1 RNA, but this is not needed for the effect, however this "bonus" binding (that is enabled by heteromultimerization) appears to enhance the retrotransposition frequency? Do you think L1TD1 is binding L1 RNA in this context or simply "stabilizing" ORF1P (Trimer) RNP?

Based on our data, L1TD1 associates with L1 RNA and interacts with L1 ORF1p. Both features might contribute to the enhanced retrotransposition frequency. Interestingly, the L1TD1 protein shares with its ancestor L1 ORF1p the non-canonical RNA recognition motif and the coiled-coil motif required for the trimerization but has two copies instead of one of the C-terminal domain (CTD), a structure with RNA binding and chaperone function. We speculate that the presence of an additional CTD within the L1TD1 protein might thereby enhance the RNA binding and chaperone function of L1TD1/ORF1p heteromultimers.

(5) Figure 2C fold enrichment for L1TD1 and ARMC1 is a bit difficult to fully appreciate. A 100 to 200-fold enrichment does not seem physiological. This appears to be a "divide by zero" type of result, as the CT for these genes was likely near 40 or undetectable. Another qRT-PCR based approach (absolute quantification) would be a more revealing experiment. This is the validation of the RIP experiments and the presentation mode is specifically developed for quantification of RIP assays (Sigma Aldrich RIP-qRT-PCR: Data Analysis Calculation Shell). The unspecific binding of the transcript in the absence of L1TD1 in DNMT1/L1TD1 DKO cells is set to 1 and the value in KO cells represents the specific binding relative the unspecific binding. The calculation also corrects for potential differences in the abundance of the respective transcript in the two cell lines. This is not a physiological value but the quantification of specific binding of transcripts to L1TD1. GAPDH as negative control shows no enrichment, whereas specifically associated transcripts show strong enrichement. We have explained the details of RIPqRT-PCR evaluation in Materials and Methods (page 14) and the legend of Figure 2C in the revised manuscript.Response: Thank you for the clarification and additional information in the manuscript.(6) Is it possible the L1TD1 antibody binds L1 ORF1p? This could make Figure 2D somewhat difficult to interpret. Some validation of the specificity of the L1TD1 antibody would remove this concern (see minor concern below).See response to (3).Response: Thanks.(7) Figure S4A and S4B: There appear to be a few unusual aspects of these figures that should be pointed out and addressed. First, there doesn't seem to be any ORF1p in the Input (if there is, the exposure is too low). Second, there might be some L1TD1 in the DKO (lane 2) and lane 3. This could be non-specific, but the size is concerning. Overexposure would help see this.The ORF1p IP gives rise to strong ORF1p signals in the immunoprecipitated complexes even after short exposure. Under these conditions ORF1p is hardly detectable in the input. Regarding the faint band in DKO HAP1 cells, this might be due to a technical problem during Western blot loading. Therefore, the input samples were loaded again on a Western blot and analyzed for the presence of ORF1p, L1TD1 and beta-actin (as loading control) and shown as separate panel in Suppl. Figure S4A.The enhanced image is clearer. Thanks.S4A and S4B now appear to the S6A and S6B, is that correct? (This is due to the addition of new S1 and S2, but please verify image orders were not disturbed).

Yes, the input is shown now as a separate panel in Suppl. Figure S6A.

(8) Figure S4C: This is related to our previous concerns involving antibody cross-reactivity. Figure 3E partially addresses this, where it looks like the L1TD1 "speckles" outnumber the ORF1p puncta, but overlap with all of them. This might be consistent with the antibody crossreacting. The western blot (Figure 3C) suggests an upregulation of ORF1p by at least 23x in the DKO, but the IF image in 3E is hard to tell if this is the case (slightly more signal, but fewer foci). Can you return to the images and confirm the contrast are comparable? Can you massively overexpose the red channel in 3E to see if there is residual overlap? In Figure 3E the L1TD1 antibody gives no signal in DNMT1/L1TD1 DKO cells confirming that it does not recognize ORF1p. In agreement with the Western blot in Figure 3C the L1 ORF1p signal in Figure 3E is stronger in DKO cells. In DNMT1 KO cells the L1 ORF1p antibody does not recognize all L1TD1 speckles. This result is in agreement with the Western blot shown above in Figure R4B and indicates that the L1 ORF1p antibody does not recognize the L1TD1 protein. The contrast is comparable and after overexposure there are still L1TD1 specific speckles. This might be due to differences in abundance of the two proteins.Response: Suggestion: Would it be possible to use a program like ImageJ to supplement the western blot observation? Qualitatively, In figure 3E, it appears that there is more signal in the DKO, but this could also be due to there being multiple cells clustered together or a particularly nicely stained region. Could you randomly sample 20-30 cells across a few experiments to see if this holds up. I am interested in whether the puncta in the KO image(s) is a very highly concentrated region and in the DKO this is more disperse. Also, the representative DKO seems to be cropped slightly wrong. (Please use puncta as a guide to make the cropping more precise)

As suggested by the reviewer we have quantified the signals of 60 KO cells and 56 DKO cells in three different IF experiments by ImageJ. We measured a 1.4-fold higher expression level of L1 ORF1p in DKO cells. However, the difference is not statistically significant. This is most probably due to the change in cell size and protein content during the cell cycle with increasing protein contents from G1 to G2. Western blot analysis provides signals of comparable protein amounts representing an average expression levels over ten thousands of cells. Nevertheless, the quantification results reflect in principle the IF pictures shown in Figure 3E but IF is probably not the best method to quantify protein amounts. We have also corrected Figure 3E.

(9) The choice of ARMC1 and YY2 is unclear. What are the criteria for the selection?ARMC1 was one of the top hits in a pilot RIP-seq experiment (IP versus input and IP versus IgG IP). In the actual RIP-seq experiment with DKO HAP1 cells instead of IgG IP as a negative control, we found ARMC1 as an enriched hit, although it was not among the top 5 hits. The results from the 2nd RIP-seq further confirmed the validity of ARMC1 as an L1TD1interacting transcript. YY2 was of potential biological relevance as an L1TD1 target due to the fact that it is a processed pseudogene originating from YY1 mRNA as a result of retrotransposition. This is mentioned on page 6 of the revised manuscript.Response: Appreciated!(10) (P16) L1 is the only protein-coding transposon that is active in humans. This is perhaps too generalized of a statement as written. Other examples are readily found in the literature.Please clarify.We will tone down this statement in the revised manuscript.Response: Appreciated! To further clarify, the term "active" when it comes to transposable elements, has not been solidified. It can span "retrotransposition competent" to "transcripts can be recovered". There are quite a few reports of GAG transcripts and protein from various ERV/LTR subfamilies in various cells and tissues (in mouse and human at least), however whether they contribute to new insertions is actively researched.(11) In both the abstract and last sentence in the discussion section (P17), embryogenesis is mentioned, but this is not addressed at all in the manuscript. Please refrain from implying normal biological functions based on the results of this study unless appropriate samples are used to support them.Much of the published data on L1TD1 function are related to embryonic stem cells [3- 7].Therefore, it is important to discuss our findings in the context of previous reports.Response: It is well established that embryonic stem cells are not a perfect or direct proxies for the inner cell mass of embryos, as multiple reports have demonstrated transcriptomic, epigenetic, chromatin accessibility differences. The exact origin of ES cells is also considered controversial. We maintain that the distinction between embryos/embryogenesis and the results presented in the manuscript are not yet interchangeable. An important exception would be complex models of embryogenesis such as embryoids, (or synthetic/artificial embryo models that have been carefully been termed as such so as to not suggest direct implications to embryos). https://www.nature.com/articles/ncb2965
https://link.springer.com/article/10.1007/s00018-018-2965-y

https://www.cell.com/developmental-cell/abstract/S1534-5807(24)00363-0?_returnURL=https%3A%2F%2Flinkinghub.elsevier.com%2Fretrieve%2Fpii%2FS1534580724003630%3Fshowall%3Dtrue

We have deleted the corresponding paragraph in the Discussion.

(12) Figure 3E: The format of Figures 1A and 3E are internally inconsistent. Please present similar data/images in a cohesive way throughout the manuscript. We show now consistent IF Figures in the revised manuscript.Response: ThanksMinor:In general:Still need checking for typos, mostly in Materials and Methods section; Please keep a consistent writing style throughout the whole manuscript. If you use L1 ORF1p, then please use L1 instead of LINE-1, or if you keep LINE-1 in your manuscript, then you should use LINE-1 ORF1p.

A lab member from the US checked again the Materials and Methods section for typos. We keep the short version L1 ORF1p.

(1) Intro:- Is L1Td1 in mice and Humans? How "conserved" is it and does this suggest function? Murine and human L1TD1 proteins share 44% identity on the amino acid level and it was suggested that the corresponding genes were under positive selection during evolution with functions in transposon control and maintenance of pluripotency [8].- Why HAP1? (Haploid?) The importance of this cell line is not clear.HAP1 is a nearly haploid human cancer cell line derived from the KBM-7 chronic myelogenous leukemia (CML) cell line [9, 10]. Due to its haploidy is perfectly suited and widely used for loss-of-function screens and gene editing. After gene editing cells can be used in the nearly haploid or in the diploid state. We usually perform all experiments with diploid HAP1 cell lines. Importantly, in contrast to other human tumor cell lines, this cell line tolerates ablation of DNMT1. We have included a corresponding explanation in the revised manuscript on page 5, first paragraph.- Global methylation status in DNMT1 KO? (Methylations near L1 insertions, for example?)The HAP1 DNMT1 KO cell line with a 20 bp deletion in exon 4 used in our study was validated in the study by Smits et al. [11]. The authors report a significant reduction in overall DNA methylation. However, we are not aware of a DNA methylome study on this cell line. We show now data on the methylation of L1 elements in HAP1 cells and upon DNMT1 deletion in the revised manuscript in Suppl. Figure S1B.Response: Looks great!(2) Figure 1:- Figure 1C. Why is LMNB used instead of Actin (Fig1D)?We show now beta-actin as loading control in the revised manuscript.- Figure 1G shows increased Caspase 3 in KO, while the matching sentence in the result section skips over this. It might be more accurate to mention this and suggest that the single KO has perhaps an intermediate phenotype (Figure 1F shows a slight but not significant trend).We fully agree with the reviewer and have changed the sentence on page 6, 2nd paragraph accordingly.- Would 96 hrs trend closer to significance? An interpretation is that L1TD1 loss could speed up this negative consequence.We thank the reviewer for the suggestion. We have performed a time course experiment with 6 biological replicas for each time point up to 96 hours and found significant changes in the viability upon loss of DNMT1 and again significant reduction in viability upon additional loss of L1TD1 (shown in Figure 1F). These data suggest that as expected loss of DNMT1 leads to significant reduction viability and that additional ablation of L1TD1 further enhances this effect.Response: Looks good!- What are the "stringent conditions" used to remove non-specific binders and artifacts (negative control subtraction?)Yes, we considered only hits from both analyses, L1TD1 IP in KO versus input and L1TD1 IP in KO versus L1TD1 IP in DKO. This is now explained in more detail in the revised manuscript on page 6, 3rd paragraph.(3) Figure 2:- Figure 2A is a bit too small to read when printed.We have changed this in the revised manuscript.- Since WT and DKO lack detectable L1TD1, would you expect any difference in RIP-Seq results between these two?Due to the lack of DNMT1 and the resulting DNA hypomethylation, DKO cells are more similar to KO cells than WT cells with respect to the expressed transcripts.- Legend says selected dots are in green (it appears blue to me). We have changed this in the revised manuscript.- Would you recover L1 ORF1p and its binding partners in the KO? (Is the antibody specific in the absence of L1TD1 or can it recognize L1?) I noticed an increase in ORF1p in the KO in Figure 3C.Thank you for the suggestion. Yes, L1 ORF1p shows slightly increased expression in the proteome analysis and we have marked the corresponding dot in the Volcano plot (Figure 3A).- Should the figure panel reference near the (Rosspopoff & Trono) reference instead be Sup S1C as well? Otherwise, I don't think S1C is mentioned at all.- What are the red vs. green dots in 2D? Can you highlight ERV and ALU with different colors?We added the reference to Suppl. Figure S1C (now S3C) in the revised manuscript. In Figure 2D L1 elements are highlighted in green, ERV elements in yellow, and other associated transposon transcripts in red.Response: Much better, thanks!- Which L1 subfamily from Figure 2D is represented in the qRT-PCR in 2E "LINE-1"? Do the primers match a specific L1 subfamily? If so, which? We used primers specific for the human L1.2 subfamily.- Pulling down SINE element transcripts makes some sense, as many insertions "borrow" L1 sequences for non-autonomous retro transposition, but can you speculate as to why ERVs are recovered? There should be essentially no overlap in sequence.In the L1TD1 evolution paper [8], a potential link between L1TD1 and ERV elements was discussed:"Alternatively, L1TD1 in sigmodonts could play a role in genome defense against another element active in these genomes. Indeed, the sigmodontine rodents have a highly active family of ERVs, the mysTR elements [46]. Expansion of this family preceded the death of L1s, but these elements are very active, with 3500 to 7000 speciesspecific insertions in the L1-extinct species examined [47]. This recent ERV amplification in Sigmodontinae contrasts with the megabats (where L1TD1 has been lost in many species); there are apparently no highly active DNA or RNA elements in megabats [48]. If L1TD1 can suppress retroelements other than L1s, this could explain why the gene is retained in sigmodontine rodents but not in megabats."Furthermore, Jin et al. report the binding of L1TD1 to repetitive sequences in transcripts [12]. It is possible that some of these sequences are also present in ERV RNAs.Response: Interesting, thanks for sharing- Is S2B a screenshot? (the red underline).No, it is a Powerpoint figure, and we have removed the red underline.(4) Figure 3:- Text refers to Figure 3B as a western blot. Figure 3B shows a volcano plot. This is likely 3C but would still be out of order (3A>3C>3B referencing). I think this error is repeated in the last result section.- Figure and legends fail to mention what gene was used for ddCT method (actin, gapdh, etc.).- In general, the supplemental legends feel underwritten and could benefit from additional explanations. (Main figures are appropriate but please double-check that all statistical tests have been mentioned correctly).Thank you for pointing this out. We have corrected these errors in the revised manuscript.(5) Discussion:- Aluy connection is interesting. Is there an "Alu retrotransposition reporter assay" to test whether L1TD1 enhances this as well?Thank you for the suggestion. There is indeed an Alu retrotransposition reporter assay reported be Dewannieux et al. [13]. The assay is based on a Neo selection marker. We have previously tested a Neo selection-based L1 retrotransposition reporter assay, but this system failed to properly work in HAP1 cells, therefore we switched to a blasticidin based L1 retrotransposition reporter assay. A corresponding blasticidin-based Alu retrotransposition reporter assay might be interesting for future studies mentioned in the Discussion, page 11 paragraph 4 of the revised manuscript.(6) Material and Methods :- The number of typos in the materials and methods is too numerous to list. Instead, please refer to the next section that broadly describes the issues seen throughout the manuscript.Writing style(1) Keep a consistent style throughout the manuscript: for example, L1 or LINE-1 (also L1 ORF1p or LINE-1 ORF1p); per or "/"; knockout or knock-out; min or minute; 3 times or three times; media or medium. Additionally, as TE naming conventions are not uniform, it is important to maintain internal consistency so as to not accidentally establish an imprecise version.(2) There's a period between "et al" and the comma, and "et al." should be italic.(3) The authors should explain what the key jargon is when it is first used in the manuscript, such as "retrotransposon" and "retrotransposition".(4) The authors should show the full spelling of some acronyms when they use it for the first time, such as RNA Immunoprecipitation (RIP).(5) Use a space between numbers and alphabets, such as 5 μg. (6) 2.0 × 105 cells, that's not an "x".(7) Numbers in the reference section are lacking (hard to parse).(8) In general, there are a significant number of typos in this draft which at times becomes distracting. For example, (P3) Introduction: Yet, co-option of TEs thorough (not thorough, it should be through) evolution has created so-called domesticated genes beneficial to the gene network in a wide range of organisms. Please carefully revise the entire manuscript for these minor issues that collectively erode the quality of this submission. Thank you for pointing out these mistakes. We have corrected them in the revised manuscript. A native speaker from our research group has carefully checked the paper. In summary, we have added Supplementary Figure S7C and have changed Figures 1C, 1E, 1F, 2A, 2D, 3A, 4B, S3A-D, S4B and S6A based on these comments.Response: Thank you for taking these comments on board!